# A full-body transcription factor expression atlas with completely resolved cell identities in *C. elegans*

Yongbin Li [1,6], Siyu Chen[2,6], Weihong Liu[2,3,6], Di Zhao[2,4,6], Yimeng Gao[1,6], Shipeng Hu[1], Hanyu Liu[1], Yuanyuan Li[5], Lei Qu [5] & Xiao Liu [1] ✉

Invariant cell lineage in *C. elegans* enables spatiotemporal resolution of transcriptional regulatory mechanisms controlling the fate of each cell. Here, we develop RAPCAT (Robust-point-matching- And Piecewise-affine-based Cell Annotation Tool) to automate cell identity assignment in three-dimensional image stacks of L1 larvae and profile reporter expression of 620 transcription factors in every cell. Transcription factor profile-based clustering analysis defines 80 cell types distinct from conventional phenotypic cell types and identifies three general phenotypic modalities related to these classifications. First, transcription factors are broadly downregulated in quiescent stage Hermaphrodite Specific Neurons, suggesting stage- and cell type-specific variation in transcriptome size. Second, transcription factor expression is more closely associated with morphology than other phenotypic modalities in different pre- and post-differentiation developmental stages. Finally, embryonic cell lineages can be associated with specific transcription factor expression patterns and functions that persist throughout postembryonic life. This study presents a comprehensive transcription factor atlas for investigation of intra-cell type heterogeneity.

While cell typing has historically relied on cellular phenotype (whether molecular, morphological, functional, physiological, or developmental), single-cell RNA-seq (scRNA-seq) transcriptome capture with unsupervised or marker-guided clustering to organize cells into biologically meaningful groups[1] is revolutionizing our definition and perspective of cell types[2]. For cell-type catalogs based on molecular profile to be meaningful, their relevance of transcriptional program to cellular phenotypes needs to be established and verified.

Phenotypes of *C. elegans* cells have been well characterized at single-cell resolution, including the invariant cell lineage throughout development, a full catalog of function- and morphology-based cell types, and a schematic of wiring for the entire nervous system[3–7].

However, accurate and comprehensive whole-body determination of individual cell identities by transcriptional profile is a long-standing challenge and non-trivial undertaking[8]. Previous scRNA-seq studies in *C. elegans* have generated relatively thorough molecular atlases of embryogenesis and the adult nervous system, respectively[9–11]. However, in these molecular atlases, the number of resolved transcriptomes represents approximately half of the total cells in each investigated system, but are considered robust largely due to transcriptomic homogeneity among symmetric cell pairs[9,11].

Alternatively, the stereotypical cell lineage and body plan of *C. elegans* can be leveraged to develop methods for identifying any (and all) given cells in a three-dimensional (3D) image stack to profile

[1]College of Life Sciences, Capital Normal University, Beijing 100048, China. [2]School of Life Sciences, Tsinghua University, Beijing 100084, China. [3]Intelligent Perception Lab, Hanwang Technology Co., Ltd, Beijing 100193, China. [4]Tianjin Key Laboratory of Exercise Physiology and Sports Medicine, Institute of Sport, Exercise & Health, Tianjin University of Sport, Tianjin 300381, China. [5]Ministry of Education Key Laboratory of Intelligent Computation & Signal Processing, Information Materials and Intelligent Sensing Laboratory of Anhui Province, School of Electronics and Information Engineering, Anhui University, Hefei 230039, China. [6]These authors contributed equally: Yongbin Li, Siyu Chen, Weihong Liu, Di Zhao, Yimeng Gao. ✉e-mail: liux@cnu.edu.cn

reporter expression for genes of interest[12–15]. A growing body of evidence suggests that batteries of transcription factors (TF) are responsible for controlling cell type identity[16–19], such as terminal selectors functioning as master regulators of cell identity[20]. Using different image analysis tools, image-based TF profiles with fully resolved cell identities have been generated covering embryogenesis[21,22], 67% cells of L1-stage larvae[23], or all neurons in adult worms[24]. However, no image analysis tool is available to annotate all cells, including fully differentiated and functional cells, across whole post-embryonic worm body.

In this work, we applied a topology-preserved matching model to automatically annotate all 558 cells in 3D image stacks of L1 larvae, then used image analysis to profile reporter expression for two-thirds of worm TFs. In our TF profiles, we found numerous instances of differential TF expression within cell types defined by cellular phenotypes. The observed intra-type heterogeneity reflected molecular mechanisms responsible for discordance among phenotypic modalities, such as morphology, function, and developmental states. Moreover, a significant source of this intra-type molecular heterogeneity was convergence of different lineages into the same phenotypic cell type, i.e., multiple cell lineages could produce cells of the same morphology and function. Indeed, the effect of embryonic cell lineage on TF expression and cell differentiation can persist after the completion of larval development. Altogether, this study presents a molecular atlas with unprecedented scope and resolution and illustrates the utility of multimodal cell typing.

## Results

### Automated cell annotation in image stacks of L1 larvae

In order to develop an automated tool for accurate, high throughput annotation of cells in 3D image stacks in *C. elegans* research, we collected 3D image stacks of 100 DAPI-stained, newly hatched L1 larvae using confocal microscopy (Supplementary Fig. 1A) and manually annotated the cell identities of all 558 nuclei in each stack as training data (Supplementary Fig. 1B, C). For convenience, we referred to different nuclei in a syncytium as separate cells. These 100 annotated image stacks were then used to as training worms to build digital worm templates. Specifically, one training worm was selected as the initial target and all straightened worm stacks were registered to the target using global affine transformation, resulting in a template L1 cell organization (Supplementary Fig. 1D–G). This process was repeated 100 times, with each training stack used once as the initial target for template generation, resulting in 100 digital templates of cell organization for L1 worms. Each digital worm template appeared as a straight rod composed of 558 cell points. Each cell point had a 3D coordinate expected in a typical L1 larva with three spatial variations along the X, Y and Z axes (Supplementary Fig. 1E, G). The template that best matched the 100 training image stacks was one derived from image stack #2 (Supplementary Fig. 2A). This template was considered the optimal digital template.

During generation of the optimal template, each worm training image stack other than stack #2 was deformed by global affine transformation. A consensus reference atlas with minimal position bias towards worm image stack #2 could be valuable for cell annotation. We, therefore followed a three-step process in which each worm training stack was first aligned to the optimal template (i.e., template #2) by global affine transformation. Subsequently, we computed the average deformation field from these transformed stacks and inverted it. This inverted parameter was then utilized to deform the optimal template, resulting in a new template. Finally, step one was reiterated and the new template replaced the optimal template. This iterative process continued until the template converged into a stable template, which was designated as the consensus template.

We then developed the Robust-Point-Matching- And Piecewise-affine-based Cell Annotation Tool (RAPCAT) algorithm to automate matching of the 558 cells in a new image stack to the corresponding cell points in reference template (Fig. 1). Given a new image stack straightened and segmented by the computation pipeline CellExplorer[12], RAPCAT computed the centroid of each cell based on segmentation masks (Fig. 1A) and fit the cell centroids to the reference template through a Robust Point Matching (RPM) approach (Fig. 1B)[25,26]. Using these initial results of RPM-based annotation, cell positions were then re-mapped through global and piece-wise affine transformation in a topologically preserved manner to minimize their displacement to the reference template (Fig. 1C) and subsequently re-assigned cell identities through bipartite matching[27] to ensure that the re-mapped cell positions and spatial distribution best fit the reference template (Fig. 1D). By default, RAPCAT ran three iterations from affine transformation through bipartite matching to optimize the cell annotations (Fig. 1C, D). After annotation, RAPCAT provided Cell Annotation Confidence (CAC) scores for every assigned cell identity, representing the likelihood of correct cell annotation. We thus defined Worm Annotation Confidence (WAC) score as the mean of CAC scores per worm.

To validate the RAPCAT annotation process, we generated 100 new image stacks of L1 larvae which were then automatically annotated by RAPCAT using either the optimal or the consensus template as reference and manually annotated in parallel. The manual annotations were used as a gold standard to test the accuracy of RAPCAT annotation, and to verify WAC score predictions of annotation accuracy. Using a 0.995 WAC score as cutoff for bipartite-improved annotation, the 100 test stacks were classified as either high or low confidence, with the two groups showing dramatically different distributions of annotation accuracy (Fig. 1E). In the high WAC group, the accuracy of RAPCAT annotation was highly correlated with WAC score and increased in many cases after the bipartite improvement. By contrast, in the low WAC group, the bipartite improvement resulted in less accurate annotation (Fig. 1E). Thus, in practice, RAPCAT outputs a bipartite-improved annotation if the WAC score for the image stack passed the 0.995 cutoff, otherwise output the RPM-based annotation. In the above test, the consensus template showed lower accuracy than the optimal template (95.3% vs. 95.9%), and fewer worm stacks passed WAC threshold (85 vs. 90) (Supplementary Fig. 2C). Based on these results, we retained the optimal template rather than the consensus template in further analyses.

Using the optimal template as reference, 10 out of the 100 testing image stacks had RAPCAT annotation accuracy rates below 90% (Fig. 1E), potentially due to a relatively poor fit between the optimal template and these 10 worms. To test whether these 10 worms can fit some of the 99 non-optimal templates better, RAPCAT re-annotated these 10 low annotation accuracy worms using each of the 99 non-optimal templates. We found that four non-optimal templates provided higher accuracy annotations (Supplementary Fig. 2B) even though their cell positions were only slightly different from those in the optimal template (Supplementary Notes). Using each of these four templates, seven of the ten test stacks passed the 0.995 WAC cutoff with mean accuracy ranging from 95.5% to 96.8% (Supplementary Fig. 2B). Based on the above results, these four alternative templates were combined with the optimal template in RAPCAT annotations of experimental worm image stacks throughout the rest of the study, i.e., RAPCAT annotates a stack using all five templates separately, and outputs the annotation results with the highest WAC score. We tested the accuracy of the 5-template-based annotation using ten reporters whose expression has been unambiguously and correctly identified in adult nervous systems and observed that the 5-template-based annotation achieved an accuracy ranging from 91.4% to 96.4% (Supplementary Notes).

The accuracy of cell identity assignments and their correlation with CAC scores were inconsistent across cells in the 100 test stacks (Supplementary Fig. 3A). So based on the 100 test stacks, we

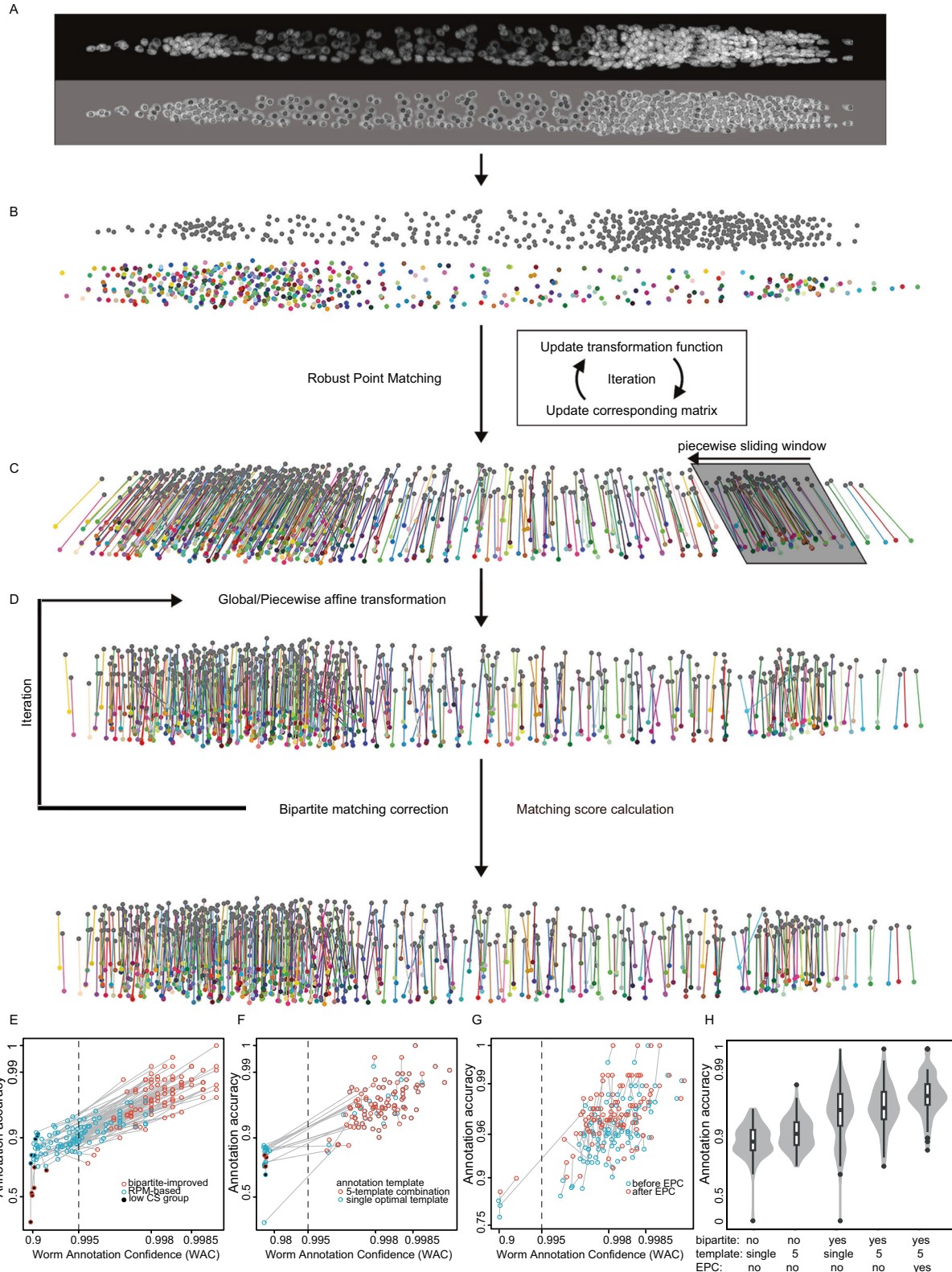

determined cell-specific CAC thresholds, below which the accuracy of cell identity assignment was less than 50% (Supplementary Fig. 3B). To improve annotation accuracy, these cell-specific CAC thresholds were used to develop an Error Prediction and Correction (EPC) option which flagged cells that did not meet their respective CAC thresholds and were then manually curated using the graphical user interface VANO[28]; RAPCAT accepted the manual curations and re-assigned the identities of other cells. To test the effectiveness of the 5-template combination

and the EPC option, we generated and analyzed another set of 100 testing image stacks. The 5-template-based annotation outperformed the optimal-template-based results both in the number of worm stacks surpassing the 0.995 WAC threshold (97 vs. 83) and in mean accuracy rate (95.4% vs. 93.1%) (Fig. 1F, H). Inspired by the superiority of the 5-template-based over the single-template-based annotation, we annotated these testing stacks using a 100-template strategy, i.e., RAPCAT annotates a stack using all 100 templates separately, and outputs the

**Fig. 1 | Automatic cell annotation by the RAPCAT algorithm. A** Extraction of 558 nuclei centroids from a new worm stack, with a 3D Z-projected slice shown for visualization. Nuclei were highlighted and segmented based on DAPI (gray) staining. **B** Initial cell identity assignment using RPM (Robust Point Matching), where the gray point set indicates a new image stack and the colored set denotes a worm template. The local point distribution of cells in the worm template is not shown. **C** 3D affine transformation of the image stack with colored lines connecting corresponding cells between the stacks, utilizing RPM-based initial annotation. **D** Bipartite improvement of cell annotation. The bipartite matching step updates cell identity assignment to maximize the overall matching score calculated by anisotropic Gaussian distribution between the worm image stack and the worm template. **E–G** Accuracy of cell annotation and prediction power of WAC score at different modes of RAPCAT. Cell identity assignment by RAPCAT was compared with that of manual annotation to calculate annotation accuracy. In each plot, two annotation modes of same image stack were linked by a gray line. The vertical

dashed lines represent the WAC cutoff to identify image stacks whose RAPCAT annotation result is highly reliable. **E** Correlation of annotation accuracy and WAC score when RAPCAT used the optimal worm template. Show are the first set of 100 testing image stacks annotated either only by the RPM (RPM-based) or after improvement by affine transformation and bipartite (bipartite-improved). Low WAC group refers to image stacks whose RPM-based and bipartite-improved WAC scores both do not pass the cutoff. CS, Confidence Score. Effectiveness of 5-template combination (**F**) and the EPC (Error Prediction and Correction) option (**G**) of RAPCAT. Shown are the second set of 100 testing worm stacks. Our RAPCAT annotated them using the 5-template combination and EPC parameters that the first set of 100 testing stacks established. Orphan points were worm stacks with identical annotations resulted from two different modes. **H** Cell annotation accuracy of the second set of 100 testing stacks at different modes of RAPCAT. Source data are provided as a Source Data file.

annotation results with the highest WAC score. However the 100-template-based annotation did not result in higher accuracy cell identity assignment than the 5-template-based (Supplementary Fig. 2D), but was markedly more time-consuming due to the greater computational burden, leading us to abandon this strategy.

On average, RAPCAT flagged 12.6 cells per worm stack that fell below their cell-specific confidence thresholds, with 5.4 of these flagged cells, on average, representing annotation errors (Supplementary Figure. 3C). After manual curation of these flagged cells, RAPCAT re-assigned cell identities resulting in a mean accuracy rate of 97.0%, i.e., 16.5 average incorrect cell identity assignments per worm (Fig. 1G, H). One round of EPC usually costs a well-trained annotator about a quarter of an hour. As a comparison, manual annotation of all 558 cells in an image stack usually costs more than 2 h.

In short, we developed RAPCAT for high accuracy, automated annotation of all cell identities in 3D image stacks of L1 larvae to facilitate analysis of large collections of worm image stacks.

### In situ TF expression profile with single-cell resolution

Based on an inclusive definition, 934 TFs are encoded in the worm genome[29]. Through the assistance of the worm research community, we obtained a collection of fluorescence protein reporter strains for 234 TFs. We then generated reporter strains for 452 TFs, in which reporter proteins were fused with histone for nuclear localization. In total, our collection spanned 657 TFs in *C. elegans* (Supplementary Data 1). Reporters of 620 TFs exhibited detectable fluorescence activity in newly hatched L1 larvae, including 864 promoter-reporter strains generated by bombardment transformation, 14 promoter reporters produced by mos1-mediated Single Copy Integration (mosSCI), 41 fosmid transgene TF protein fusion reporter strains, and 23 Cas9-mediated TF knock-in strains (Supplementary Data 1).

These strains were DAPI stained and confocal scanned to generate a total of 1055 3D image stacks (Supplementary Data 1), 301 of which had been manually annotated during RAPCAT development as described above (Supplementary Data 1). The remaining 754 stacks were computationally straightened and segmented by the CellExplorer pipeline[12]. Our worm collection protocol had a chance to collect larvae a little older than the early L1 stage so that their Q neuroblasts had divided one round and migrated. We first manually examined Q neuroblasts of these segmented stacks and identified 103 larvae little older than the early L1 stage (Supplementary Data 1), which were manually annotated because they did not well fit our digital template trained by early-stage L1 larvae. The remaining 663 stacks were automatically annotated by RAPCAT annotation process, including the EPC option. Among these image stacks, 27 did not pass the 0.995 WAC cutoff and were thus manually annotated. To estimate the accuracy of RAPCAT annotation of the 636 stacks, we manually examined 76 stacks (Supplementary Data 1) and observed a mean accuracy rate of 95.7% and

high correlation between WAC score and annotation accuracy (Supplementary Fig. 3D).

The resulting identities of cell centroids were transferred to corresponding cell masks in the image stacks using VANO[28], which returned reporter signal data for each mask to generate the expression profile of each TF across all 558 cells (Supplementary Data 2). Excluding Z2/Z3 germ cells from expression profiling because of their transgene silencing[30] (Fig. 2A), the final data included 620 TF expression profiles across all 556 somatic cells in L1 larvae (Supplementary Data 2). Multiple worms were imaged for 99 of the reporter strains, more than half of which had correlation coefficients of $R > 0.81$ for reporter expression patterns between individual worms of the same strain (Fig. 2B). These results indicated that the nuclei annotations were robust and, importantly, the reporter expression was reproducible. In addition, expression profiles of 234 TF reporter constructs were obtained from multiple worm strains (i.e., with reporter integration at different sites). Comparison between different strains showed that reporter expression patterns were less similar among image stacks of the same reporter construct in different strains (median of $R = 0.71$) than among different image stacks from the same strain (Fig. 2B), suggesting that differences in transgene integration site among worm strains could lead to differences in reporter expression patterns.

It is well-established that promoter-fusion reporters may not always fully recapitulate the expression patterns of their corresponding genes if the necessary regulatory elements are located outside the upstream sequence used in the construct, such as those in introns[31,32] or distal enhancers in long intergenic regions[33]. Similarly, the transcription of downstream genes in an operon is usually controlled by a promoter outside of its direct upstream intergenic sequence[33], and is consequently excluded from screens of the immediate upstream region, such as our promoter fusions. We therefore tested the extent of their influence by comparing the profiles of 84 TFs shared between our L1 profile and the previously published adult profiles[24]. This set included 58 promoter-fusion reporters, 22 fosmid-based reporters, and 4 knock-in reporters (Supplementary Data 1), with fosmid and knock-in reporters serving as gold standards because they contained the full genomic context of the respective genes[24].

Target genes were then classified into four types, i.e., upstream intergenic sequence >7 kb, upstream intergenic sequence <1 kb, genes with >1 kb intron following the start codon, and all other genes. The cloned promoter regions of genes categorized in the fourth group were presumed to contain the necessary and sufficient regulatory elements for driving their expression, termed as high-context promoter reporters. Substantiating this proposition, the L1 profiles of these high-context promoter reporters displayed strong correlations (median of ROCAUC = 0.92) with corresponding adult TF profiles[24]. This correlation level resembled the relationship observed between the gold standard fosmid/knock-in reporters and the adult profiles

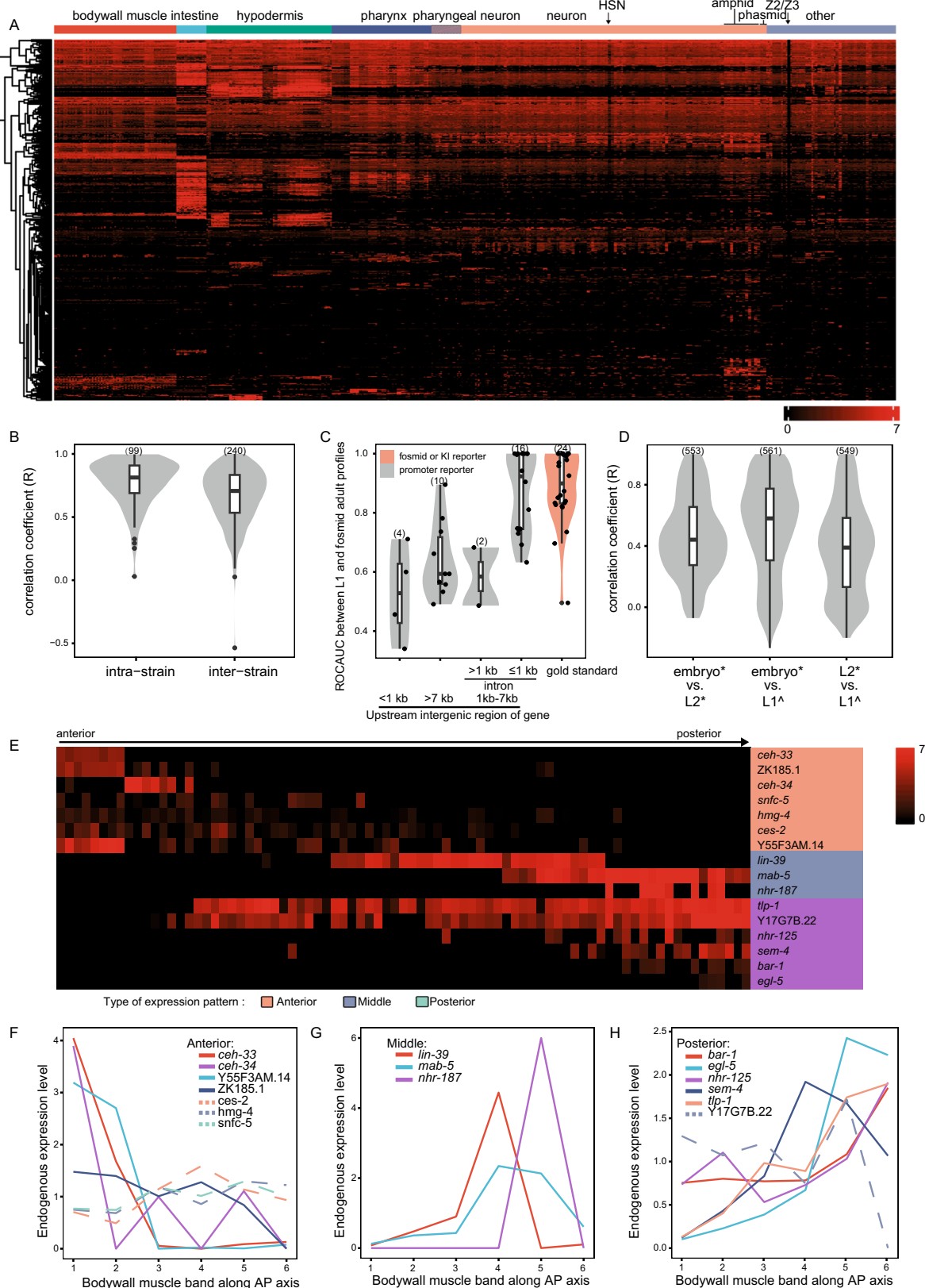

**F** Anterior: *ceh-33*, *ceh-34*, Y55F3AM.14, ZK185.1, *ces-2*, *hmg-4*, *snfc-5*

**G** Middle: *lin-39*, *mab-5*, *nhr-187*

**H** Posterior: *bar-1*, *egl-5*, *nhr-125*, *sem-4*, *tlp-1*, Y17G7B.22

(median of ROCAUC = 0.90) (Fig. 2C). In our study, these high-context promoter reporters constituted a total of 296 TFs. In contrast, promoter fusions belonging to the first three categories, referred to as low-context promoter reporters, displayed weak correlations with corresponding adult TF profiles[24] (Fig. 2C). These low-context promoter reporters encompassed 263 TFs (Supplementary Data 1).

These reporter expression profiles were then compared with previously published scRNA-seq datasets of late embryos[11] and L2 larvae[8]. In total, 58 independent cell types in L1 larvae were identified in both scRNA-seq analyses (Supplementary Data 3). Comparison of TF expression profiles across these shared cell types showed that their embryonic scRNA-seq profiles were more correlated with their

**Fig. 2 | Whole body TF expression profile at single-cell resolution. A** Expression atlas of TF reporters with full-solved cell identity. Genes were clustered according to their expression profiles. Cells were manually arranged according to their cell types. **B** The correlation of TF profiles among worms originating from same reporter transgene strain, as well as among reporter transgene strains derived from the same reporter construct. **C** The correlation between our profiles in L1 larvae and previously published reporter data in L4/young adults. A dot is a reporter whose ROCAUC was computed based on our profile in L1 larvae and the previously published expression data of the corresponding gene's fosmid reporter in L4/young adults[24]. The TFs were categorized based on the genomic context of their respective reporters. intron, the intron right downstream of the first coding exon. **D** The correlation between our reporter results and previously published scRNA-seq data. The previously published scRNA-seq data did not have all cell identities resolved[8–11]. To make two types of data comparable, cells resolved in scRNA-seq data were grouped into non-overlapping cell types shared by all three datasets. **E** Heterogeneous gene expression in bodywall muscles along anteroposterior axis in the TF reporter atlas. Genes are arranged according to their expression patterns. Gene expression patterns along bodywall muscle bundles are depicted at right side of the heatmap. The 81 bodywall muscles are arranged along their anteroposterior positions. Lineage-dependent TFs in Fig. 6B was excluded to avoid confounding effect. **F–H** Endogenous expression patterns of TFs whose reporters showed anteroposterior expression in (**E**). Bodywall muscles are grouped into six bands along anteroposterior axis as previously reported[11]. Gene expression is represented by log2 size-factor normalized UMI counts in the scRNA-seq data[11] and normalized by mean. Genes whose expression patterns are not consistent with their reporter profile are in dotted lines. Numbers in parentheses indicate strains or genes in (**B–D**). Source data are provided as a Source Data file.

reporter profiles in newly hatched L1 larvae than with their scRNA-seq profiles of L2 larvae (Fig. 2D), which was consistent with the stability of the histone-reporter fusion protein[22] and with the pervasive transcriptomic changes observed in post-mitotic cells during larval development[34]. Among TFs showing differential expression along the anteroposterior bodywall muscle bundles in our reporter screen (Fig. 2E), 16 TFs were detected in bodywall muscles in the scRNA-seq studies[8,11]. 75% of these TFs displayed anteroposterior expression patterns that were consistent between their reporter profiles and scRNA-seq data (Fig. 2F–H). Taken together, our in situ reporter profiles closely matched scRNA-seq data from adjacent developmental stages, but with completely resolved cell identity.

Another concern is the different turnover rates of endogenous TFs and our histone::mCherry reporter protein. We generated two knock-in reporter strains of *vab-15*, a protein fusion reporter and a 2A::histone::mCherry co-expression reporter. The protein fusion reporter could capture the expression dynamics of endogenous VAB-15 protein, whereas the 2A reporter co-expressed fluorescence reporter and target gene by ribosomal skip mechanisms during translation so that reporter protein and target protein were generated as independent molecules[35]. Consistent with the high stability of histone protein, fluorescent signal of the 2A co-expression vector was observed in all cells detected by the VAB-15 fusion reporter as well as in dozens of additional cells, such as G2 neuroblasts (Supplementary Fig. 4F). By comparison, a previously published scRNA-seq study in worm embryos[11] found that *vab-15* mRNA could be detected at levels slightly above background in G2 neuroblasts. Moreover, genetic analysis below revealed that *vab-15* contributed to G2 proliferation potential, thus confirming the endogenous expression of *vab-15* gene in G2 neuroblasts. These collective results illustrate the advantage of a histone::mCherry reporter in detecting weak gene expression[22].

The expression patterns of these two *vab-15* knock-in reporters also differed in V5 neuroblasts (Supplementary Fig. 4F). Interestingly, both reporters were positive in Q neuroblasts, the sister cells of V5 neuroblasts. In late embryos, *vab-15* is transcribed in the mother cells of V5 and Q neuroblasts[11], and specifies the fates of V5 and Q cells[36]. In L1 larvae, the development of Q neuroblasts still requires regulation by *vab-15*, while V5 development does not[36]. Q and V5 cells are generated by the last round of embryonic cell division, which occurs only an hour before embryo hatching[7]. So the *vab-15* expression observed in V5 cells of L1 larvae via the 2A co-expression reporter is likely a false positive resulting from the long half-life of histone::mCherry reporter protein along with the short duration from division of the mother cells to early L1 stage.

Fortunately, all other embryonic cell divisions occur more than 6 h before embryo hatching[7], which is likely long enough to ensure that all of the histone reporter protein is degraded. To test this hypothesis, we compared reporter expression between daughter cells and their mother cells. In total, 126 histone::mCherry reporters were profiled at the L1 stage in the current study and in embryos up to the 350-cell stage at single cell resolution[22], including 216 mother cells whose daughter cells were both present in L1 larvae. Among them, 207 mother cells showed strong expression (>30,000 intensity units) of at least one reporter. In 129 of these mother cells, fluorescence of at least one reporter with strong signal in the mother became undetectable in one or both daughter cells of L1 larvae (Supplementary Data 5). Moreover, daughters of these 129 mother cells spanned all five major tissues (intestine, muscle, neural cell, pharynx, and skin), suggesting that histone fusion reporter proteins produced in mother cells were completely degraded in most, if not all, somatic cells prior to worm fixation and imaging.

## Phenotype vs. TF profiling in cell classification

In *C. elegans*, cell types have been defined with varying granularity based on their phenotypic features[4–6]. In this study, the term coarse cell type is used to define 25 broad cell types (e.g., neuron, neuroblast, and pharyngeal muscle) that are each comprised of multiple sub-types or cell classes that differ in morphology and/or function, while the cell-class is used to represent 146 high-resolution cell types composed of phenotypically homogeneous cells with no distinct sub-classes (e.g., intestine, bodywall muscle, and intestinal muscle) (Supplementary Data 6).

We searched our TF profiles for absolute cell type specificity, defined as expression in all members of one cell type, but not in any cell of other cell types. No TF in our profiling was absolutely specific for any coarse cell type. By contrast, 12 TFs showed absolute specificity in seven cell classes, including six TFs (*ada-2*, *elt-2*, *ets-9*, *nhr-121*, *nhr-176*, and R09H10.3) in the intestine; *pes-1* in the Z1/Z4 somatic gonad precursor; *npax-4* in the phasmid sheath; *ceh-44* in IL2 neurons; *pqm-1* in LUA neurons; and *che-1* and *nhr-74* in amphid neurons ASE and AWA, respectively (Supplementary Data 4). The large number of intestine-specific TFs supports the notion that endoderm is a special tissue in the transcriptome[37]. Our profiles included five known intestine regulators (*elt-2*, *elt-7*, *end-1*, *end-3*, and *med-1*)[38]. All but *med-1* were expressed in all 20 intestinal cells in our profiling.

Unlike intestine, no specific TF was detected in bodywall muscles in our profiles. Bodywall muscles are specified by three redundant master regulators, *hlh-1*, *unc-120*, and *hnd-1*[39]. Each of these regulators is expressed in multiple cell types. Nevertheless, two major master regulators, *hlh-1* and *unc-120*, were co-expressed exclusively in all 81 bodywall muscles (Supplementary Data 2), distinguishing them molecularly from all other cell classes. Moreover, every two cell classes could be distinguished in our profiling in that at least one TF was expressed in all members of one cell class, but not in any cell of the other cell class (Supplementary Data 4). Altogether, the substantial heterogeneity and coverage of our TF expression data suggested that close examination of the concordance between phenotype- and TF expression-based cell classification schemes was warranted.

We generated a dendrogram by hierarchical clustering of our TF profiles that revealed 76% of the 144 phenotypic cell types composed

of multiple cells were distributed as single clades (Fig. 3), supporting a high degree of concordance between phenotypic and molecular modalities. Moreover, the high degree of concordance could retain even after slightly decreasing the number of TFs, cell annotation accuracy, or faithfulness of reporters to endogenous genes (Supplementary Note), illustrating high coverage and quality of our expression data. However, dozens of phenotypic cell types were distributed across multiple clades (Fig. 3, Supplementary Data 6), illustrating discrepancies between phenotype-based classification and molecular clustering.

To investigate how transcriptionally divergent cells shared similar phenotypes, we conducted molecular subtyping within these multiclade cell types. Briefly, clusters of cells belonging to same type in separate clades were defined as subtypes (Supplementary Fig. 6A). Cells of other phenotypic types that shared the most similar TF profile with that of a given subtype were defined as its neighbor-clade cells (Supplementary Fig. 6B). This process ultimately classified these multiclade phenotypic cell types into 129 clades as their TF-defined subtypes, 72 of which were distinct from any conventional subtypes classified by cellular phenotype (Fig. 3, Supplementary Data 6). Next, Jensen Shannon Divergence (JSD) values were calculated using TF expression profiles to quantify the intra-type transcriptional differences between these subtypes. The TF profile-based subtypes with the highest intra-type JSD scores occurred in neuron, glial cell, hypodermis, pharyngeal muscle, and neuroblast (Supplementary Fig. 6G). We then characterized the discrepancies between phenotype- and TF profile-based classifications in these cell types.

### Neuron and glial subtypes express different numbers of TFs

All neurons were classified by TF profiling into an overall neuron subtype, only excluding HSNs, amphid and phasmid neurons (Fig. 4A). Similarly, all glial cells were classified into an overall glial subtype except amphid, and phasmid glial cells (Fig. 4B). The amphid sensillum is the largest chemosensory organ, while the phasmid sensillum is structurally similar, but smaller[4]. The TF profiles showed that amphid neurons expressed significantly more TFs than other neurons (Fig. 4C). Moreover, sheath cells in the amphid and phasmid sensilla expressed significantly more TFs than those not in these two sensilla (Fig. 4C). Comparison with previously published scRNA-seq data for embryos and L2 stage larvae[8] verified our reporter-based observations that significantly more TFs were expressed in amphid and phasmid cells than in other neurons or glial cells (Fig. 4D, E).

Another TF-based subtype overlapped completely with phenotypic HSN neuron class (Fig. 4A). During embryogenesis, HSN neurons persist as quiescent round cells after the threefold stage[7], with HSN neurite outgrowth initiating at the L1 stage and completing by the end of larval development[40]. So, TF profiling in the current study was conducted at the end of HSN neuron quiescence, resulting in their consistent classification as a phenotypically and transcriptomically distinct cell group[41]. Among all TFs profiled here, 86% were expressed at their lowest levels across all somatic cells in the L1 stage (Figs. 2A, 4F). Moreover, our profiles included three protein fusion reporters (ceh-38, lin-13, and snu-23) whose expression was detected in nearly all neurons. The expression level of each of these reporter in HSN class was the bottom-2 among all 96 neuron classes (Supplementary Fig. 4G). One potential explanation for this effect was the broad downregulation of TFs specific to the HSN quiescent state. To test this hypothesis, we examined TF expression in HSNs at the 1.5-fold embryo stage, during which HSNs are developmentally active[7], and at the young adult stage, in which HSNs are functional[40].

To this end, we randomly selected 11 TFs with various reporter types that are broadly expressed in adult worms. All of these TFs were expressed in HSNs at levels comparable to their neighbor cells (ALM, hypodermis and vulva) at the young adult stage (Supplementary Table 1). Then two reporters (saeg-2 mosSCI and sma-9 knock-in) were

examined in detail (Fig. 4G–I and Supplementary Fig. 5). Expression of saeg-2 occurs in newly formed HSNs, becoming undetectable at the 3-fold stage (Fig. 4I and Supplementary Fig. 5A, B). HSNs had similar sma-9 expression levels as their neighbor neurons at the threefold stage (Fig. 4I and Supplementary Fig. 5F). At the L1 larval stage, both saeg-2 and sma-9 expression was barely detectable (Fig. 4I and Supplementary Fig. 5C and G). We found saeg-2 expression is initially restored in the L2/L3 stages, while sma-9 expression remains undetectable (Fig. 4I and Supplementary Fig. 5D, H). In young adults, both genes were expressed in HSNs at levels equivalent to that in other neurons (Fig. 4I and Supplementary Fig. 5E, I). Overall, these analyses thus show that TF expression is temporally repressed in quiescent HSNs during post-embryonic differentiation. At last, these two reporters remained repressed in HSNs of ced-3 L1 larvae of both genders (Supplementary Fig. 5J–M), suggesting that HSN repression was gender-independent and not related to apoptosis.

### Concordance between TF profiling and morphology

Characterization of TF-based subtyping of hypodermis, pharyngeal muscle and neuroblast suggested that subtype classification based on TF profiling is consistent with intra-type morphological heterogeneity (Supplementary Notes). Moreover, although a TF-based subtypes are different cell types than their respective neighbor clades, they share remarkably similar morphology (Supplementary Notes). For example, neuroblast at the L1 stage have various epithelial morphology, but proliferate to produce neural progeny during larval development[6,7]. Our TF-based subtyping identified four neuroblast subtypes, including P, G1/G2/W, Q/V5/T, and K. For each subtype, its neighbor clade was comprised of epithelial cells with highly similar morphology (Fig. 5A). Moreover, the smallest inter-subtype difference in gene expression was found between the K and G1/G2/W neuroblast subtypes (Fig. 5B), consistent with their similar epithelial morphology. So, we speculated that partitioning intra-subtype molecular difference by stratification of morphological heterogeneity could help identify which neuroblast subtypes share similar pro-neural TF batteries (Supplementary Notes) (Supplementary Fig. 6D).

The smallest normalized JSD score was found between the P and G2/W neuroblast subtypes (Fig. 5C). Four regulators of P neuroblasts have been reported[36], three of which are expressed in G2/W cells (ref-2, vab-15, and tlp-1) (Fig. 5A). In ref-2 knockdown worms, both G2 (Fig. 5E, L) and W (Fig. 5H, M) neuroblasts both gave rise to fewer progeny than in wild type. On the contrary, no phenotype of either G2 or W neuroblast was observed in tlp-1 null mutants (Fig. 5I and 5L-5M). Knocking down vab-15 in wild-type background, ref-2 was downregulated strongly in the G2 neuroblast (Fig. 5F and L), but only slightly in the W neuroblast (Fig. 5J, M). However, knocking down vab-15 in tlp-1 null background, strong ref-2 downregulation occurred in the W neuroblast (Fig. 5K, M). But the synergistic effect between vab-15 and tlp-1 was not observed in the G2 neuroblast (Fig. 5L).

In summary, TF expression in neuroblasts is more related to their morphology than their proliferative potentials. Stratification of TF profiles by normalizing JSD score with subsequent genetic analysis, revealed a TF battery (ref-2, vab-15, and tlp-1) shared between the TF-based G2/W and P neuroblast subtypes.

### Postembryonic TF relevance to embryonic lineage expression

In light of our findings that TF expression can vary strikingly between subtypes of different cellular phenotypes, we next examined variation in TF expression within phenotypically homogenous cell classes. Returning to our dendrogram of 181 clusters, each comprised of cells with statistically indistinguishable TF profiles (Fig. 3, Supplementary Data 6), we found that 49% TF profile clusters shared complete overlap with phenotypic cell types while 47% clusters partially overlapped with only one phenotypic cell class, and were thus designated as TF-defined subclasses (Supplementary Data 6). The

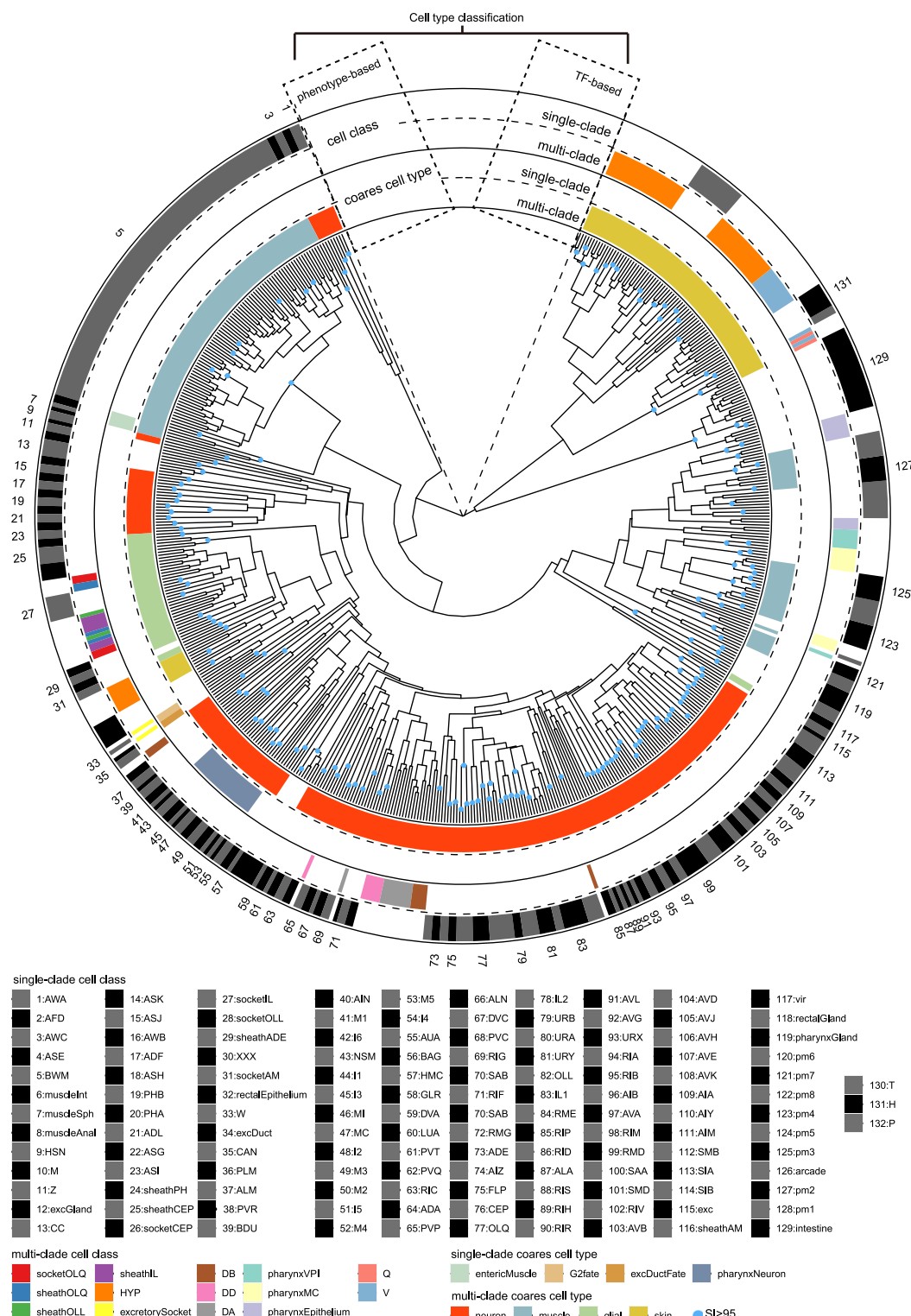

**Fig. 3 | Hierarchical clustering of the 556 somatic cells by reporter expression.** Surrounding the circular dendrogram are phenotypic cell types of the 556 somatic cells in four concentric layers. Cell classes are in the outer bi-layer while coarse cell types in the inner bi-layer. Within a bi-layer, cell types overlapping completely with single clades of the dendrogram are in the outer layer while those distributed in multiple clades are in the inner layer. Single-clade cell classes are numbered clockwise to facilitate lookup. Due to space limitation, only odd numbers are labeled in the circular dendrogram. SI, selective inference support value as percent. Source data are provided as a Source Data file.

large number of TF-defined subclasses illustrated the higher resolution of TF profiling for cell type classification than that provided by phenotype. Within phenotypic cell classes, differences in embryonic cell lineage distance between cells within a subclass were significantly shorter than those between cells in different subclasses (Paired $t$ test, two-sided, $P$ value $< 10^{-16}$) (Fig. 6A), indicating that cell lineage was correlated with TF expression profile.

However, it should be noted that cells in closely related lineages commonly occupy the same or close locations, spatial expression patterns can confound lineage determination. We, therefore, focused

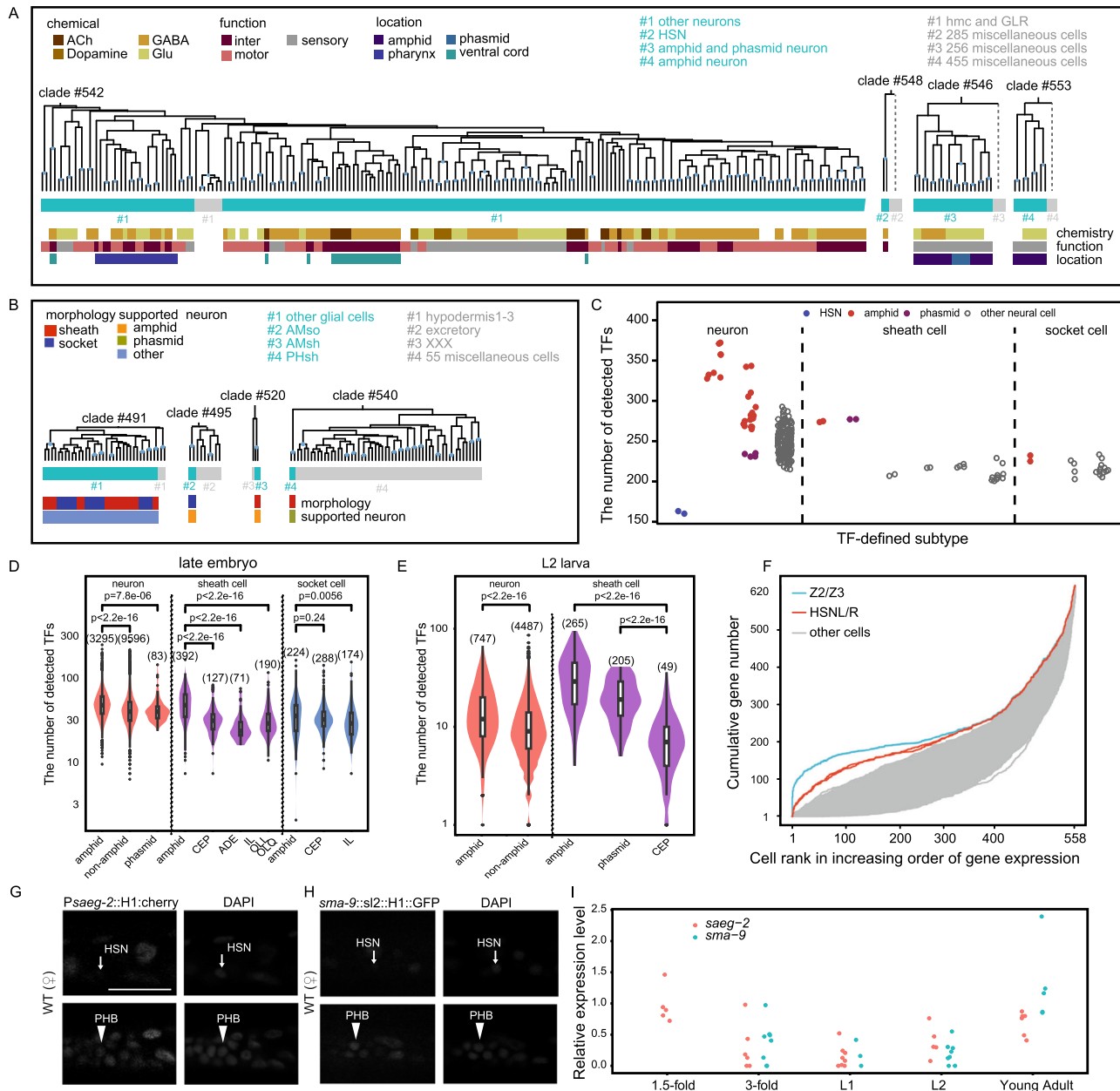

**Fig. 4 | Subtyping of neural cell types by the TF profiles.** Clades from the dendrogram showing TF-defined subtypes of two phenotypic cell types, i.e., neuron (**A**) and glial cell (**B**). At the bottom of each panel are conventional cell types based on various phenotypic features. Number sign in blue, TF-defined subtype; Number sign in gray, neighbor-clade cells. A hashed vertical line, discarded clade of numerous cells to save space. **C** TF expression in every neural cell according to the 620-TF profiles. At each X-axis point are cell members of a TF-defined subtype of neuron, sheath cell, or socket cell. TF expression in single neural cells detected by previous scRNA-seq studies[8,11] at late embryo stage (**D**) and L2 larval stage (**E**). Cell numbers are indicated in parentheses. *P* value was calculated based on the

Wilcoxon test, two-sided. *P* values are indicated directly in the figure. **F** Enrichment of low expression TFs in each cell identity. Cell ranks are in the order of increasing gene expression level based on the 620-TF profiles. **G**–**I** Expression of *saeg-2* and *sma-9* reporters in HSNs. **G**, **H** Left-side of L1 larvae are shown. Sister cells PHB and HSN neurons were photographed with the same exposure time. Ten (**G**) and five (**H**) animals were scored, respectively (Scale bar, 10 µm). **I** Temporal reporter expression in HSN neurons. Expression level is represented by the ratio of reporter fluorescence in HSNs over that in neighbor neurons. Animals of *sma-9* reporter were not scored at 1.5-fold stage. Source data are provided as a Source Data file.

on cells with variations in TF expression that did not align with any spatial patterns, such as the ventral bodywall muscle bundles, where MS-derived and C-/D-derived cells are intercalated[7]. We detected 18 lineage-dependent TFs in the ventral bodywall muscle bundles (ten MS-enriched, three C-exclusive, four D-exclusive, and one MS-exclusive TFs; Fig. 6B). Notably, two of these 18 TFs (MS-enriched *hmg-11* and MS-exclusive *unc-130*) still retained their lineage-dependent expression patterns in anterior bodywall muscles even after completion of larval development (Fig. 6C).

Equivalent morphology and function can be also observed in 159 bilateral cell pairs whose cell mates are located in strict symmetry at the L1 stage[6], excluding K/K' and ASE cell pairs due to their developmental and neural activity asymmetry, respectively[6,42]. It is well-established that 97 of these 159 cell pairs are symmetric in cell lineage, while the other 62 pairs are converged cell pairs, whose left and right cell mates have an asymmetric cell lineage history[7]. For example, the intestino-rectal valve (vir) cell pair are symmetric in cell lineage in that its left mate and right mate are sister cells. On the contrary, intestinal

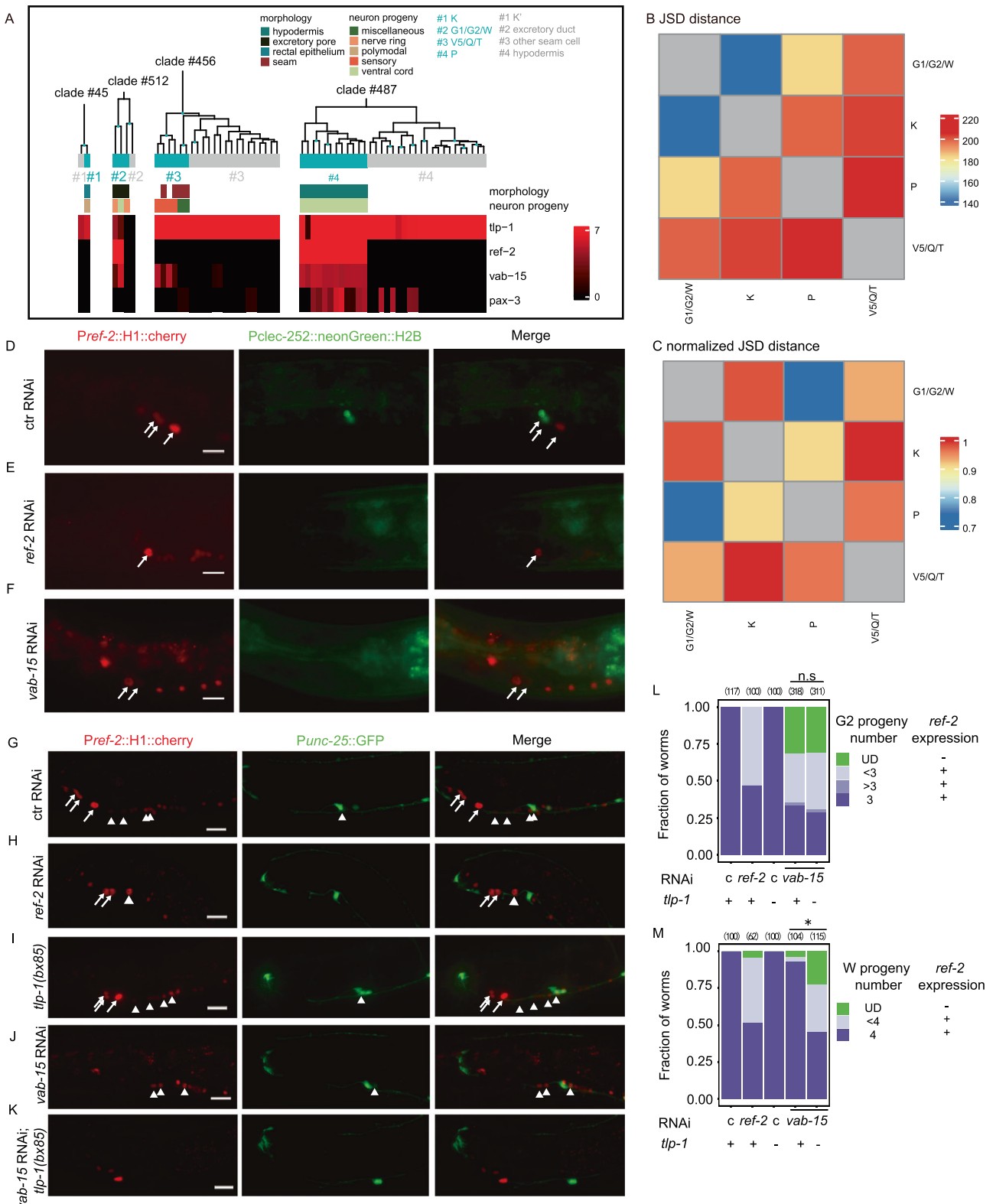

muscle (im) cell pair are asymmetric in cell lineage in that its left mate (imL/AB.plpppppaa) right mate (imR/MS.ppaapp) are derived from distinct blastomeres[7]. In our L1 stage TF profiles, qualitatively asymmetric TF expression was significantly more enriched in the convergent pairs than those derived from symmetric lineage (24/62 vs. 1/97; Fisher's exact test, two sided, *P* value = 2.6e-08) (Fig. 6D). In young adults, most of these 13 TFs lost expression asymmetry. Nevertheless, three TFs (*nhr-67*, *ceh-27*, and *ref-1*) retained their asymmetric

expression in the intestinal muscle pair, I1 neuron pair, IL2 and RMDD neuron pairs, separately (Fig. 6E).

One plausible explanation of these observed persistent lineage-specific expression is long half-life of histone::mCherry reporter protein. However, development from L1 to young adult takes about two days, much longer than the half-life of histone:: mCherry reporter protein as described above. For example, histone:: mCherry reporters of six TFs (*med-1*, *hlh-2*, T22C8.4, *zip-7*, *tbx-8*, and *tbx-9*) were expressed

**Fig. 5 | Regulators of post-embryonic cell lineage of G2/W neuroblasts. A** TF-defined subtypes of neuroblast. Number sign in blue, TF-defined subtype; Number sign in gray, neighbor-clade cells. The W neuroblast is categorized as having excretory pore morphology because it becomes G2 in case of G2 ablation[7]. Expression profiles of four well-known regulators of the P neuroblasts are shown. Distance in TF expression between TF-defined subtypes of neuroblast before (**B**) and after (**C**) normalization by cross-neighbor clade distance. JSD, Jensen Shannon Divergence score. Names of TF-defined subtypes are their cell class members, instead of their number signs in (**A**). Phenotyping post-embryonic development of G2 (**D–F**) and W (**G–K**) neuroblasts in negative control (**D, G**), *ref-2* knockdown (**E, H**), *vab-15* knockdown (**F, J, K**), and *tlp-1* null deletion (**I, K**). Every experiment was repeated twice. The P*ref-2*::H1::mCherry reporter was used as a marker to label G2/W progeny. P*clec-252*::neoGreen::H2B and P*unc-25*::GFP reporters were used to label G2-derived RMF neurons[9] (**D–F**) and W-derived D-type ventral cord motor neuron (**G–K**)[36], respectively. Each worm was arranged such that its anterior end was to the right and its ventral midline was at the bottom. All of the worms scored were at L3 stage. There was no W progeny in photos of (**D–F**) because these photos were taken more anteriorly than those of (**G–K**). Arrow, G2 or its progeny; arrowhead, W or its progeny. (Scale bar, 10 µm) (**L, M**) Phenotypes of proliferation and marker expression of G2 (**L**) and W (**M**) lineages. Parentheses, the number of scored worms; UD, undetermined due to undetectable expression of *ref-2* marker; c, negative control. (**L**) n.s., *P* value = 1.0 as for the fraction of worms whose W progreny lost *ref-2* expression and P-value = 0.26 as for the fraction of worms with less than four W progeny. (**M**) asterisk, *P* value = 4.0e−10[-5] as for the fraction of worms whose W progeny lost *ref-2* expression and *P* value = 1.0e-10[-14] as for the fraction of worms with less than four W progeny (Fisher's exact test, two-sided). Source data are provided as a Source Data file.

in MS-derived bodywall muscles of L1 larvae, but became undetectable in those of young adults (Fig. 6B, C). Moreover, while *ref-1* was specifically expressed in the right RMDD neuron of L1 larvae (Fig. 6D), its expression was undetectable in the neuron of late embryos[15,21], suggesting post-embryonic initiation of asymmetric expression of *ref-1*. Altogether, these data supported the significant effect of embryonic cell lineage on post-embryonic gene expression, and moreover, this lineage dependency might persist throughout post-embryonic life.

### Lineage-specific TFs in convergent differentiation

In our above reporter profiles, the intestinal muscle cell pair had the highest number of asymmetric TFs (Fig. 6D), including two TFs (*egl-38* and *nhr-67*) specific to the left intestinal muscle (imL/AB.plppppppaa), and four (F21D5.9, *irx-1*, *pal-1* and *unc-39*) specific to the right intestinal muscle (imR/MS.ppaapp). The *Six/SO*-family homeobox TF, *unc-39*, showed a typical lineage-specific expression pattern and was expressed at the L1 stage in all embryonic progeny cells of MS.(a/p)pa, including five cell classes (Fig. 6F). Two cell classes (M myoblast and coelomocyte) are only generated by the MS.(a/p)pa lineage, while the other three (the intestinal muscle, the bodywall muscle, and GLR) converge from at least two cell lineages[7]. Among these converged cell classes, only cells derived from the MS.(a/p)pa lineage expressed *unc-39* (Fig. 6F).

We traced lineage in *unc-39*-null embryos to the 350-cell stage, at which point nearly all cell divisions in the MS.(a/p)pa lineage are complete in the wild type. Patterns of MS.(a/p)pa lineage cell division in *unc-39*(*gk798*) homozygotes were indistinguishable from that in heterozygotes (Fig. S7), indicating that *unc-39* was not required to generate MA.(a/p)pa progeny. Previous analyses have shown that *unc-39* mutants are defective for the specification of the M myoblast class and the coelomocyte class[43], both of which are completely derived from the MS.(a/p)pa lineage[7]. We examined the role of *unc-39* in the specification of a convergent cell class, the intestinal muscle. In *unc-39*(*gk798*) mutants, the imR/MS.ppaapp cell did not express its marker gene *arg-1* and non-striated muscle specifier *msl-1* in *unc-39*(*gk798*) mutants, while the imL/AB.plppppppaa cell remained unaffected (Fig. 7A, B). In short, *unc-39* is a lineage-specific TF and participates in convergent differentiation of the intestinal muscle classes.

Next, we sought regulators of the imL/AB.plppppppaa cell. In our TF profiles, both intestinal muscles expressed four TFs known to be involved in the development of body muscles, including the bodywall muscle specifiers, *hnd-1* and *unc-120*[39], and the non-striated muscle specifiers, *hlh-8*[44] and *mls-1*[45] (Fig. 6F and Supplementary Data 2). Interestingly, expression of the HAND bHLH TF, *hnd-1*, is specific for the AB.p(l/r)pppppa lineage in embryos, whereas its expression is undetectable in the MS.(a/p)pa lineage as late as the comma stage, when the imR/MS.ppaapp cell has already been generated[21]. Post-embryonic worms showed remarkably stronger *hnd-1* expression in the imL/AB.plppppppaa cell than in the imR/MS.ppaapp cell at all examined stages (Fig. 6G, H, and Supplementary Fig. 8).

Both intestinal muscles still expressed *arg-1* marker genes in *hnd-1* mutants (Fig. 7E), suggesting that the fate of intestinal muscles are not specified by *hnd-1*. We then examined their morphology (Fig. 7F-J). In *hnd-1*(*q740*) mutant L1 larvae, the imL/AB.plppppppaa cell was significantly shorter than imR/MS.ppaapp (Fig. 7E, I). To investigate the post-embryonic role of *hnd-1*, we generated an *hnd-1* somatic knock-out strain in which Cas9 is driven by a heatshock promoter. After heatshock-induced disruption of *hnd-1* in newly hatched L1 larvae, the imL/AB.plppppppaa cell had significantly fewer finger-like projections than imR/MS.ppaapp (Fig. 7F, J). Moreover, disrupting *hnd-1* at the young adult stage, after completion of larval development, still led to a similar asymmetric pattern of the phenotype (Fig. 7G, J).

Altogether, these data show that the lineage-dependent TFs, *unc-39* and *hnd-1*, are expressed in a mirror image patterns relative to each other and play asymmetric roles during convergence of the intestinal muscles. Moreover, the asymmetrical expression and role of *hnd-1* persists even after larval development ends.

## Discussion

Here, we developed the RAPCAT algorithm to automate annotation of all cell identities in 3D image stacks of L1 larvae, enabling analysis of large collections of worm reporter strains. We then analyzed the reporter expression profiles of 620 TFs, accounting for two-thirds of TFs encoded by the worm genome. Their reporter expression in L1 larvae was quantified to generate an expression atlas at single-cell resolution. The TF profiles were then used to re-categorize phenotypic cell types, and phenotypic types within heterogeneous molecular clusters were further divided into subtypes based on variation in their TF expression. Finally, we characterized the contribution of transcriptome size, phenotypic modality, and cell lineage to intra-type heterogeneity in TF expression.

RNA-seq analysis generally assumes that transcriptome size is similar among different cell types[46], which was largely supported by whole-body reporter profiling in the current study. However, the observed large-scale downregulation of TF expression in HSN neurons specifically at the L1 stage is an exception to this principle. Time-series analysis suggested that gene expression is globally repressed in HSN neurons in a temporal manner consistent with the well-established quiescent state of HSNs during post-mitotic differentiation. Suppressed transcription has been observed in quiescent primordial germ cells (PGCs) of *C. elegans*, *Drosophila*, echinoderms, ascidians, and mice, with RNAPII phosphoSer2 depletion reported as a conserved molecular mechanism likely underlying this PGC transcriptomic profile[47–50]. Similar to PGCs, subsets of mammalian somatic stem cells reside in a quiescent state[51–54], although it remains unclear whether global transcriptional repression is associated with dormancy in mammalian somatic quiescent stem cells.

Integration of developmental, morphological, functional, and transcriptomic phenotypic modalities is useful for classifying cells[55,56], but such classifications are confounded by non-trivial discordance

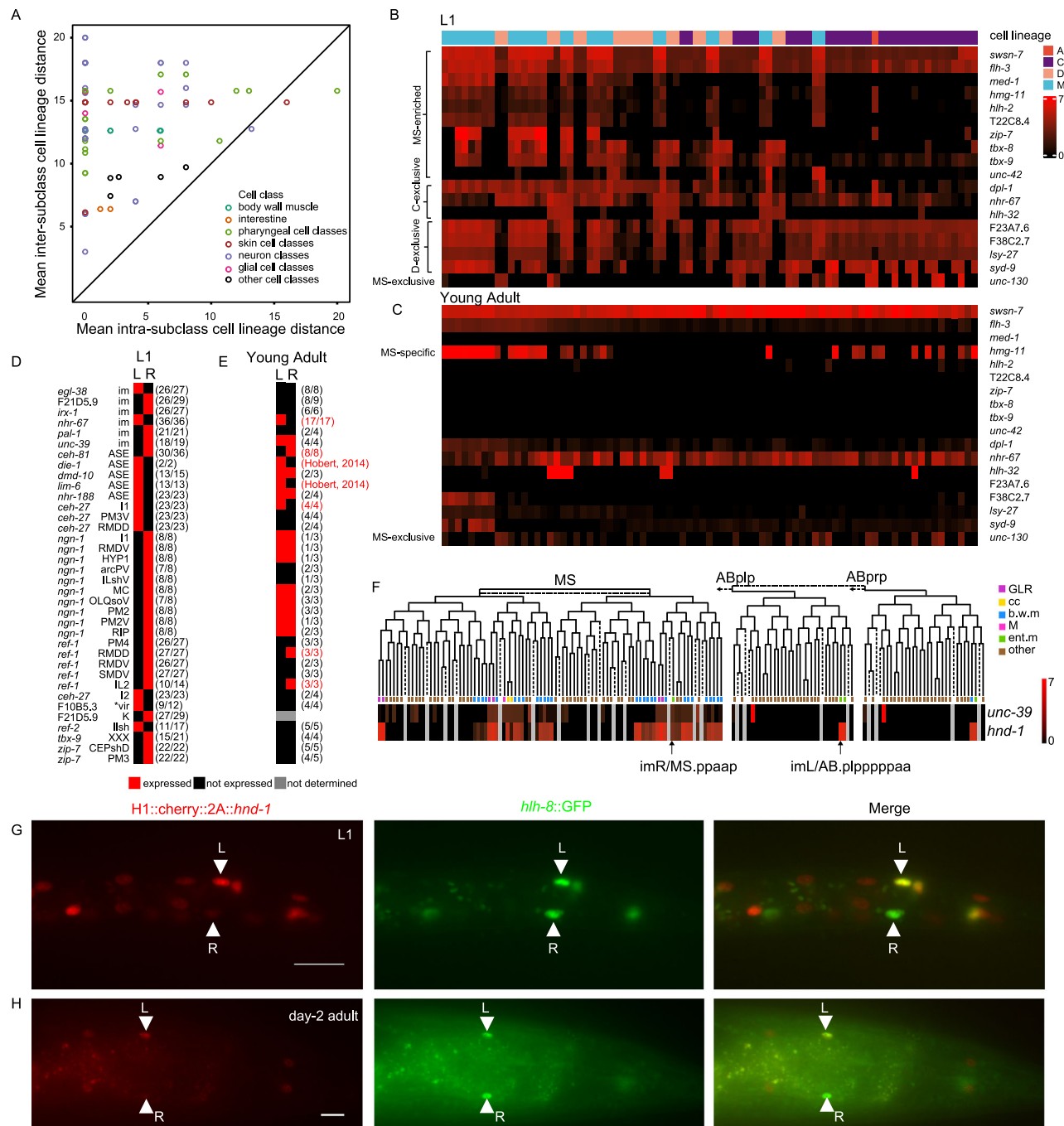

**Fig. 6 | Association of cell lineage with postembryonic TF expression patterns.**
**A** Correlation between embryonic cell lineage and TF expression profiles between TF-defined subtypes within phenotypic cell classes. Cell-lineage-related TFs in bodywall muscles (**B**, **C**) and anatomically symmetric cell pairs (**D**, **E**) at L1 stage (**B**, **D**) and at the stage of young adult stage (**C**, **E**). **B**, **C** Bodywall muscles are arranged along anteroposterior axis. M-derived muscles, not displayed here to maintain the consistency of cell positions across the two heatmaps, are presented in Supplementary Fig. 9. **D**, **E** Parentheses, the number of worms with shown expression patterns over the number of scored worms. Bodywall muscles are arranged along anteroposterior axis. M-derived muscles during larval development are not shown so the positions of cells remain unchanged in two heatmaps. Asymmetric expression patterns in adults are marked by red (**E**). Asterisk, the cell

pair with symmetric embryonic lineage. **F** Expression profiles of *unc-39* promoter reporter and *hnd-1* knockin reporter in progeny of MS and AB.p(l/r)pp lineages. Cells are arranged according to their embryonic cell lineage. Dotted lines link equivalent groups. Dashed lines represent apoptotic cells. Dotted arrows represent ABp(l/r)pa cell lineages not shown to save space. cc. coelomocyte. b.w.m, bodywall muscle. ent..m, enteric muscle. Expression patterns of *hnd-1* knockin reporter in the intestinal muscles at the L1 stage (**G**) and day-2 adult (**H**). Ventral view of posterior region of a L1 larva (**G**) and a day-2 adult (**H**) to show expression patterns of *hnd-1*. The intestinal muscles are marked with arrowheads. Worms were oriented with left to the top. 20 animals were scored for each stage. (Scale bar, 10 μm) L, imL/ AB.plppppppaa; R, imR/MS.ppaapp. The *hlh-8* protein fusion reporter was used as a marker to label intestinal muscles. Source data are provided as a Source Data file.

among these modalities[2]. Gene expression profiles are commonly integrated with phenotypic modality to resolve incongruities in cell type taxonomy[57–59]. Our analyses reveal high intra-type heterogeneity in TF expression in hypodermis, pharyngeal muscle, and neuroblast-

type cells. TF-based subtyping of these classes aligned well with the heterogeneous morphology and function displayed by these cells.

For example, analysis of developmental phenotype in neuroblasts shows that they give rise to various neurons during larval

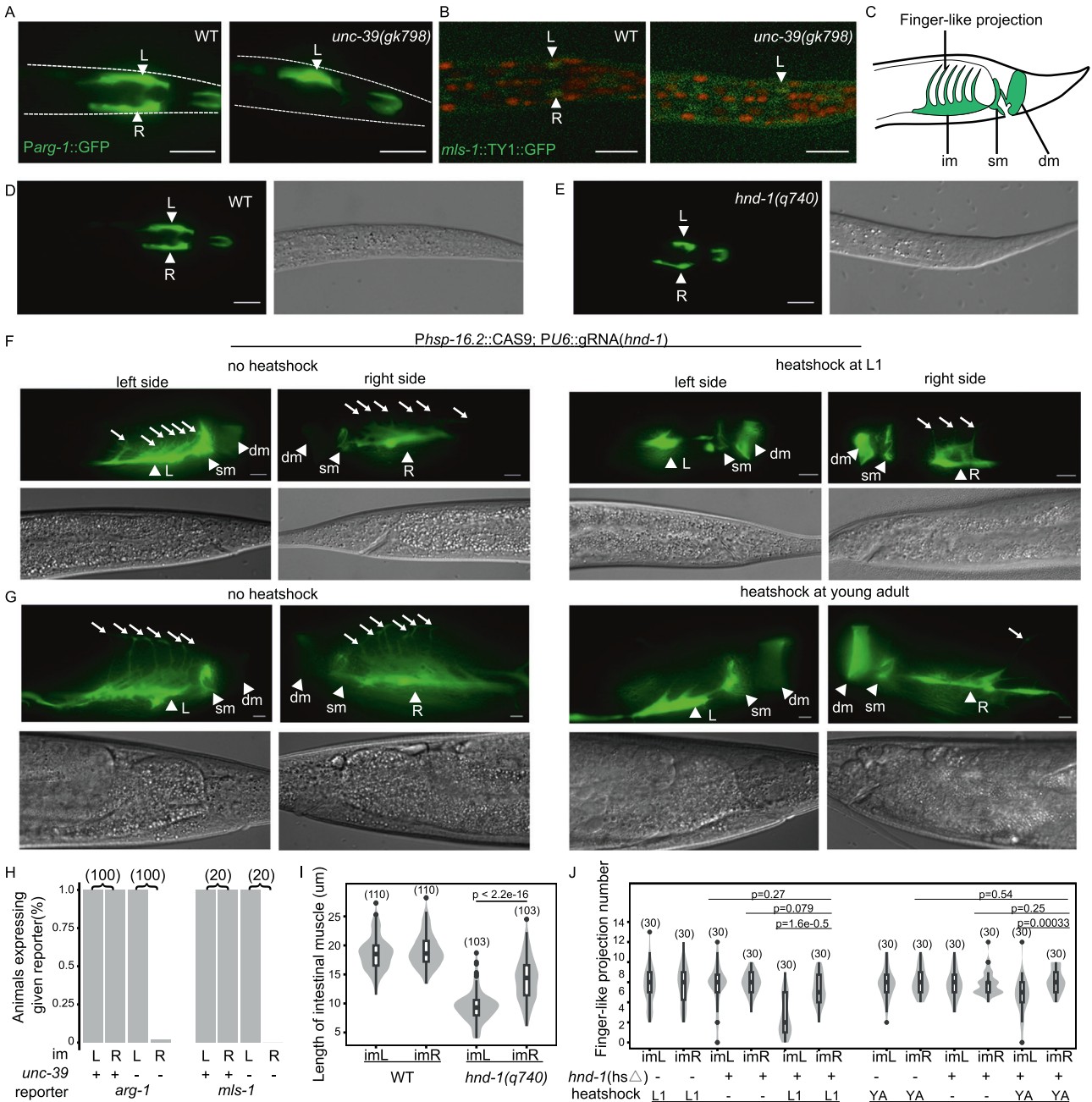

**Fig. 7 | Asymmetric requirement of lineage-specific TFs *unc-39* and *hnd-1* in development of the intestinal muscle pair.** Ventral view of posterior regions of L1 larvae to show expression patterns of *arg-1* (**A**) and *mls-1* (**B**). Worms were oriented with left to the top. L, imL/AB.plpppppaa; R, imR/MS.ppaapp. Statistics is shown in (**H**) (Scale bar, 10 μm). **C** A diagram of worm tail region showing the location and morphology of the enteric muscles, adapted from ref. 44. im, intestinal muscles; sm, anal sphincter muscle; dm, anal depressor muscle. **D**–**G** Morphology of the intestinal muscles at ventral view (**D**, **E**) and lateral view (**E**, **G**) of worms. The morphology of enteric muscles was visualized by cytoplasmic GFP driven by the *arg-1* promoter. **D**, **E** Shown are L1 larvae. Worms were oriented and intestinal muscles were labeled in the same way as in (**A**, **B**). **E**, **G** Worms were heatshocked to

knockout *hnd-1* at the L1 (**D**) and young adult (**E**) stages, 2 days before photography. Worms were oriented with dorsal to the top. Arrow, finger-like projection of intestinal muscles. Only projections that reached dorsal midline were counted. **H** Fraction of L1 larvae expressing marker genes in the intestinal muscles. **I**, **J** Distribution of morphological traits of intestinal muscles. Parentheses, the number of scored worms per marker and genotype. *P* value was calculated based on the Wilcoxon test, two-sided. The exact *P* values have been annotated within the figure. *hnd-1*(hs△), genotype P*hsp-16.2*::Cas9; PU6::gRNA(*hnd-1*) so that heatshock can activate Cas9 expression to knockout *hnd-1*; YA, young adult. Source data are provided as a Source Data file.

development, while morphological and functional phenotyping suggest that they can act as various epithelial cells in a context-dependent manner. TF-based hierarchical clustering of neuroblasts illustrates that clusters corresponding to morphology and function branch off at higher levels than those corresponding to development. Notably, W and P neuroblasts have identical neurogenic cell lineages and share a proneural TF battery, but are distantly related in the TF profile-based

hierarchy, respectively serving as neighbor-clade cells of the excretory duct cell and hypodermal cells. These results suggest that the gene regulatory programs in neuroblasts are dominated by morphology and function more than their role in development.

Widespread convergence of co-fated lineages is well-established in complete cell lineage maps of *C. elegans*[6,7], and two-layered TF cascades have been shown to regulate convergence in neurons[60], glial

cells[61], and bodywall muscles[62,63]. In this type of cascade, upstream lineage-specific TFs are transiently expressed in lineage-specific progenitor cells to establish competency for fate specification, while post-mitotically expressed are downstream type-specific terminal selectors[60–63]. Single-cell profiling in developing worm embryos has identified transient molecular correlations between cell lineage and expression profile before the final embryonic cell division[11,21], after which the expression profile changes abruptly in post-mitotic differentiating cells resulting in homogenous scRNA-seq profiles among phenotypically indistinguishable cells of different lineages[11]. This post-embryogenesis loss of correlation between expression profile and cell lineage is supported by scRNA-seq of fully differentiated neurons in adult worms, wherein members of the same neuron class from different lineages have indistinguishable transcriptomic profiles[9].

By contrast, TF profiling and genetic assays in the current work reveal that embryonic cell lineage has non-transient effects on gene expression and cell differentiation. First, embryonic cell lineage continues to play a significant role in gene expression patterns in post-embryonic cells. In several cases, lineage-dependent TF expression among phenotypically indistinguishable cells persists even after larval development is complete, such as biased expression of *hnd-1* in imL/ AB.plpppppaa. Moreover, *hnd-1* plays an asymmetric role in the morphogenesis of intestinal muscles during and even after completion of larval development. Incomplete convergence of gene expression patterns may also occur in mammals[2]. For example, scRNA-seq analyses of cultured cells of the mouse hematopoietic system detected more than a dozen of differentially expressed genes between mature monocytes derived from granulocyte-monocyte progenitors and those from monocyte-dendritic progenitors[64].

Cumulatively, the TF-based hierarchy in the current study appears to share a strong relationship with cell morphology and developmental history. It is reasonable to expect that profiling functional genes could provide insights into cell function and physiology. For example, the expression profile of genes encoding signaling proteins in the body-wall muscle bundles contribute to signal gradients along the anteroposterior axis[65] and dorsoventral axis[66]. Our RAPCAT algorithm can be trained to annotate 3D image stacks of larvae at other stages and adult worms. Obtaining gene expression data for each cell throughout development can provide a foundation for dissecting cellular phenotypes across an entire organism.

## Methods

### *C. elegans* strains and maintenance
All *C. elegans* strains used in this study are provided in Supplementary Data 1 and 7. All strains were raised on Nematode Growth Media plates and fed OP50 *E. coli*[67] at 20 °C in incubators, unless otherwise noted.

### Worm image stack acquirement and pre-processing
**Worm staining and confocal scanning.** Worms hatched within 3 h were collected as early stage L1 larvae. Worm fixation and DAPI-staining used a protocol modified from previous publication[23]. Briefly, worms were washed by M9, spun down, and then quickly re-suspended by 4% PFA in Modified Ruvkun's Witches Brew (MRWB) and frozen in liquid nitrogen overnight. Worms were thawed at 4 °C, rotating at least 2 h, washed by Tris-Triton Buffer (TTB) with 100 mM DTT for 5 min, and then stained by DAPI or Hoechst at 1 μg/mL for 3 h. Stained worms were washed by TTB for five times and mounted in 60% glycerin for microscopy. 3D image stacks of L1 larvae were obtained using a Zeiss confocal microscope with a ×63 oil objective (NA = 1.4). X-Y and Z dimension sampling was set at 0.116 μm and 0.122 μm per pixel, respectively.

**Image stack straightening and segmentation.** Confocal microscopy images were processed and converted to TIFF formats using ImageJ (v1.50_i). The image analysis pipeline CellExplorer[12] was used to computationally straighten these image stacks. Nuclei in each image stack were segmented automatically in DAPI channel[12] and then manually curated using the VANO (v1.741) interactive interface. Manual annotation or examination of nuclear identities was based on the prototypical morphology and relative spatial positions of nuclei in *C. elegans* previously described in literature[6] and the WormAtlas[68,69].

### Development of image analysis tool to automate cell annotation
**Building digital worm templates.** In the 100 manually annotated training worm image stacks, VANO was used to generate mass center of each segmented nucleus to represent corresponding cells. The 558 mass centers of each worm were considered as a set of 3D Cartesian coordinates. To build a template, we took one of the 100 training worms as the target worm ($T_i$), and all 100 worms were mapped to $T_i$ by using a 12-parameter 3D affine transformation as Eq. (1).

$$\begin{pmatrix} X \\ Y \\ Z \\ 1 \end{pmatrix}_{T_i} = \begin{pmatrix} p_1 & p_2 & p_3 & p_4 \\ p_5 & p_6 & p_7 & p_8 \\ p_9 & p_{10} & p_{11} & p_{12} \\ 0 & 0 & 0 & 1 \end{pmatrix}_{worm_j \to T_i} \begin{pmatrix} X \\ Y \\ Z \\ 1 \end{pmatrix}_{worm_j} \quad (1)$$

$$(i = 1: 100, j = 1: 100)$$

which for simplicity, can be written as Eq. (2).

$$\begin{pmatrix} X_{T_i} \\ 1 \end{pmatrix} = M_{j \to T_i} \begin{pmatrix} X_j \\ 1 \end{pmatrix} \quad (2)$$

$$(i = 1: 100, j = 1: 100)$$

Where $X_{T_i}$, $X_j$ are the position vectors of target worm $T_i$ and worm j, and $M_{j \to T_i}$ represents the transformation matrix from worm$_i$ to target worm $T_i$. Then the transformation matrix $M_{j \to T_i}$ is calculated solving above equation and the 3D affine transformed worm$_i$ is calculated as Eq. (3).

$$\begin{pmatrix} X_{j \to T_i} \\ 1 \end{pmatrix} = M_{j \to T_i} \begin{pmatrix} X_j \\ 1 \end{pmatrix} \quad (3)$$

$$(i = 1: 100, j = 1: 100)$$

A digital worm template based on target worm $t_i$ was then generated by averaging the position of each cell across 3D-affine transformed training worms as Eq. (4).

$$Template_i = \frac{\sum_{j=1}^{100} \left( X_{j \to T_i} \right)}{100} (i = 1: 100) \quad (4)$$

The standard deviation of each cell in Template$_i$ was calculated based on coordinates of specific cell among the 100 3D affine transformed training worms.

Next, we try to identify an optimal template from the 100 template candidates. Each template candidate was evaluated by piecewise affine transformation to quantify the displacement of cells in each training worm to the template candidate. Cell positions in candidate template$_i$ were used as the target point set and those in training worm$_j$ as the subject point set. We implemented piecewise affine transformation by first defining the length of piecewise sliding window as 1/8 of the total length of the candidate template$_i$, and the sliding-step as 1/8 of the length of sliding window. Then, we iteratively moved the sliding window along the AP direction of the average template with the sliding-step as step size. In each step, the cells of candidate template$_i$ within each sliding window and their corresponding cells in training

worm_j were extracted and affine aligned. Finally, the aligned positions of the same cell in all steps were averaged as the final piecewise affine alignment results. To take into account both the cell spatial variation information and relative position encoded in the template to enforce the shape prior to cell distribution during matching, we calculated the cell matching score for each cell between template_i and a piecewise-affined worm using anisotropic Gaussian distribution function as Eq. (5).

$$f_{ijk} = A \bullet \exp\left\{ -\frac{1}{2}\left[ \frac{(x_{ik} - \bar{x}_{jk})^2}{\sigma_{\bar{x}_{jk}}^2} + \frac{(y_{ik} - \bar{y}_{jk})^2}{\sigma_{\bar{y}_{jk}}^2} + \frac{(z_{ik} - \bar{z}_{jk})^2}{\sigma_{\bar{z}_{jk}}^2} \right] \right\}$$

$$(i = 1: 100, j = 1: 100, k = 1: 558)$$

Where, $x_{ik}, y_{ik}, z_{ik}$ are coordinates of cell $k$ in piecewise affine transformed worm_j, $\bar{x}_{jk}, \bar{y}_{jk}, \bar{z}_{jk}$ and $\sigma_{\bar{x}_{jk}}, \sigma_{\bar{y}_{jk}}, \sigma_{\bar{z}_{jk}}$ are coordinates and SDs of cell $k$ in template_i, respectively. We took $A$=1 to simplify calculation.

The mean of 55,800 cell matching scores for template_i represents its atlas matching score.

**Automated cell annotation.** For each new straightened and segmented worm, the 3D center coordinates of its 558 segmented nuclei were treated as subject point set and center coordinates of 558 cells in the digital were treated as target point set. The automatic annotation problem is transformed into point set registration problem between subject and target point sets.

(1) Three principal axes of target and subject point-sets were extracted using principal component analysis and rigidly aligned. (2) Following the deterministic annealing framework of RPM[25,26] we iteratively mapped the subject point-set to the target one to establish the initial matching between two point-sets. (3) We mapped the subject point-set to the target by using global affine transformation followed with piecewise affine transformation based on the initial cell matching results obtained in step 2. The length of piecewise sliding window was 1/8 of the total length of the candidate template_i, and the sliding-step was 1/8 of the length of sliding window. (4) We used bipartite matching to optimize the cell matching obtained in step 3. We modeled the matching score of each cell in the worm image stack to every cell in the digital worm template by using the anisotropic Gaussian distribution with optimized parameters as Eq. (6).

$$f_{ij} = \exp\left\{ -\frac{1}{2}\left[ \frac{(x_i - \bar{x}_j)^2}{(\alpha\sigma_{\bar{x}_j})^2} + \frac{(y_i - \bar{y}_j)^2}{(\alpha\sigma_{\bar{y}_j})^2} + \frac{(z_i - \bar{z}_j)^2}{(\alpha\sigma_{\bar{z}_j})^2} \right] \right\}$$

$$(i = 1: 558, j = 1: 558)$$

Where $x_i, y_i, z_i$ are coordinates of cell $i$ in subject worm, $\bar{x}_j, \bar{y}_j, \bar{z}_j$ and $\sigma_{\bar{x}_j}, \sigma_{\bar{y}_j}, \sigma_{\bar{z}_j}$ are coordinates and SDs of cell $j$ in the cell position template, respectively. $\alpha$ is constant that controls the overall smoothness of the field of matching score. We took $\alpha = 20$ in this study.

Next, cell identity is recognized by solving a bipartite graph matching problem using Hungarian Algorithm to maximize the overall matching score between new worm and the template. (5) Finally, our annotation goes back to step 3 with updated cell-matching results to iterate until converge to satisfying degree. In practice, we iterated three times.

**Developing indexes to predict the accuracy of automatic cell annotation.** We define the Cell Annotation Confidence (CAC) score of

cell identity assignment for every cell as Eq. (7).

$$f_i = \exp\left\{ -\frac{1}{2}\left[ \frac{(x_i - \bar{x}_i)^2}{(\alpha\sigma_{\bar{x}_i})^2} + \frac{(y_i - \bar{y}_i)^2}{(\alpha\sigma_{\bar{y}_i})^2} + \frac{(z_i - \bar{z}_i)^2}{(\alpha\sigma_{\bar{z}_i})^2} \right] \right\}$$

$$(i = 1: 558, \alpha = 20)$$

Where $x_i, y_i, z_i$ are coordinates of cell $i$ in subject worm, $\bar{x}_i, \bar{y}_i, \bar{z}_i$ and $\sigma_{\bar{x}_i}, \sigma_{\bar{y}_i}, \sigma_{\bar{z}_i}$ are coordinates and SDs of cell $i$ in the cell position template, respectively. The mean of 558 CCS in a worm image stack represents its Worm Annotation Confidence (WAC) score.

Every examined worm image stack was annotated by both manual and RAPCAT -based automatic procedures. An error was called if manual and automatic annotation assigned different identities to a cell. The number of errors ($N_{error}$) was counted for each examined image stack, and $1 - N_{error}/558$ represented the accuracy rate of each worm.

### Reporter expression profiling at single-cell resolution
The 3D image stacks were first computationally straightened and then registered into a canonical rod shape that had the same orientation and size using CellExplorer (Supplementary Fig. 4A, B). Next, CellExplorer automatically segmented image stacks to identify nuclei as bright objects in the foreground of dark (Supplementary Fig. 4C). Because the segmentation software was designed for the trunk and tail of L1 larvae[12], there are typically numerous segmentation errors in the densely packed brain region of the worm that require manual curation to resolve. Using VANO, a well-trained worm biologist may require 2 h to identify and curate all segmentation errors. An image stack without segmentation errors has around 558 nuclear masks, depending on whether there are more or less than 20 intestinal nuclei.

To measure the reporter expression level for every cell, background fluorescence was estimated using ten pseudonuclei manually generated using VANO. After subtracting background fluorescence, reporter fluorescence was normalized by DAPI fluorescence to account for spherical aberration. A nucleus with a normalized reporter fluorescence of 500 was often barely distinguishable from background fluorescence. To diminish the effect of background signal on gene expression analysis, reporter expression level was calculated as $\log_2$((normalized reporter fluorescence +500)/500) (Supplementary Fig. 4D). Different types of reporters captured the expression of the respective genes to a different extents. From high extent to low extent were fosmid/knock-in, high-context promoter, and low-context promoter. If there were different types of reporters for the same TF, the reporter type with higher faithfulness was chosen. Then, the mean expression profile across all image stacks of a strain was used as the strain expression profile. The mean expression profile across all strains of a TF was used as the expression profile of the TF. Heatmaps were generated using the ComplexHeatmap (v2.12.1) R package.

### Computation analysis of the TF profiles
**Hierarchical clustering of the 556 somatic cells based on TF profiles.** TF expression profiles were clustered using the R pvclust package (v2.2-0) with default parameters. Specifically, average linkage was used as the agglomerative method and correlation was used as the distance measure, with bootstrap 1000 and relative sample size ranging from a proportion of 0.5 to 1.4 of the original sample size. The relative proportion was incremented by 0.1 for each bootstrap resampling. The selective inference (SI) support values > 95 suggest well-supported clusters. Dendrograms were visualized using the ggplot2 (v3.4.1) and ggdendro (v0.1.23) R packages.

**Subtyping of conventional cell types based on the TF profile dendrogram.** For a cell type in the TF profile dendrogram, all cells of other cell types were assigned as not-given-type (Supplementary Fig. 6A). Branches connecting two nodes of given cell type were defined as intra-type edges, while branches connecting a node of given cell type and that of other type were considered as inter-type edges (Supplementary Fig. 6B). The subtyping pipeline used a parsimony algorithm[23] to assign given cell type to internal nodes in a manner that minimized the number of inter-type edges. A subtype was defined as a cell group whose intra-group connections were all intra-type edges and whose connections with outside nodes were all inter-type edges (Supplementary Fig. 6C). Moreover, the node connecting a subtype by the fewest inter-type edges was defined as the neighbor clade of the subtype (Supplementary Fig. 6C).

**Distance in gene expression between cell groups.** Using the philentropy(v0.7.0) R package, the Jensen Shannon Divergence (JSD) scores between cells were computed. The JSD score between two groups was the mean of JSD scores between all cell members of these two groups. The JSD score between a subtype and another subtypes' neighbor-clade was defined as cross-neighbor clade JSD. The JSD score between two subtypes over the mean of their cross-neighbor clade JSDs was defined as the normalized inter-subtype JSD score (Supplementary Fig. 6D).

## Identification of TFs with specific spatial patterns at the L1 stage based on the TF profiles

**Anteroposterior TFs in the bodywall muscles.** There are four bodywall muscle bundles along the worm body[6]. Two types of expression patterns along bodywall muscle bundles were defined as anteroposterior in bodywall muscles. The first was region-specific patterns. A 10-cell long window slid along all four muscle bundles. A TF was defined as region-specific if there was a position where every bundle had at least one cell expressing the TF at more than twofold level of background and all muscles out of the window had undetectable expression. The second pattern was gradient, in which a TF satisfied both criteria in more than one muscle bundles: (i) not expressed in one end of a muscle bundle; (ii) $P$ value $< 10^{-3}$ using lineage regression.

**Lineage-related TFs in the bodywall muscles.** The AB-derived bodywall muscle was excluded from the assay because only one bodywall muscle is derived from AB blastomere[7]. Other 80 bodywall muscles were grouped according to which blastomeres there are derived from. If all muscles in a group expresses a TF at higher levels than all muscles in another group, this TF was defined as lineag-related.

**Stereotypical asymmetric TFs in anatomically and morphologically symmetric cell pairs.** TFs whose expression was more than fourfold than background in one mate of cell pair but undetectable in another cell mate were selected as candidates for asymmetric genes. For each candidate, we examined at least eight worms. If there were at least 75% of worms showing asymmetric expression in the same direction, the TF was defined as asymmetric in this cell pair.

## Reporter strain construction

Promoter sequences were defined as the intergenic sequences upstream the start codon using the criteria as previously described[36] (Supplementary Data 2). Each promoter was cloned into pUbHG or pUbHC in frame with *his-24*::GFP or *his-24*::mCherry, respectively[15]. Transgenic worms were generated by microparticle bombardment using *unc-119* as a selection marker as reported[70]. The single-copy insertion transgenic strains were generated using MosSCI[71]. Each promoter::H1::mCherry fragment was cloned into pCFJ350, then it was injected into EG4322 worms along with PJL43.1 (Mostase coding vector). Two selection marker PCFJ90 (P*myo-2*::mCherry) and PCFJ104

(P*myo-3*::mCherry) were co-injected to help to screen integrated transgenic worms. For knockin strains, reporter sequence was inserted into the N- or C-terminus of the genomic locus of the TF using the CRISPR/Cas9 system[72]. Endogenous TF coding sequence and reporter sequence are separated by trans-splicing acceptor SL2 sequence[73] or 2A self-cleaving peptide sequence[74] (Supplementary Data 7).

## Somatic knockout

Guide RNA sequences of *hnd-1* were: 5′-ctctcttccgatttggaggta-3′; 5′-caaatgatcaatgtactgca-3′; 5′-tcgtgctcaatgtatcaact-3′. These sequences were cloned into the vector pOG2306 designed for somatic KO[75]. All there vectors mixed with the P*myo2*::mCherry, P*hlh-8*::H1::mCherry and P*rsp-27*::NeoR coinjection marker was injected into *ccIs4443*[*arg-1*::GFP+*dpy-20*(+)]. Synchronized L1 of transgenic strain were transferred to OP50-seeded nematode growth medium plates and heat shocked at 33 °C for 1 h[75]. The L1s then grew at 20 °C 2 days and only those worms with fully expressed P*hlh-8*::H1::mCherry reporters were scored for mutant phenotypes.

## RNAi knockdown

Full-length cDNAs of *vab-15* and *ref-2* were cloned to pL4440. In vitro transcription (IVT) template contain dual opposing T7 promoters were generated by PCR with primer sequences: 5′-gttttcccagtcacgacgtt-3′ and 5′-cgaggaagcaacctggctta-3′. Large scale IVT were performed with 3 μL unpurified PCR product, 2 μL T7 Polymerase (Beyotime Biotechnology), 2 μL rNTP mix (NEB), 1 μL RNAse Inhibitor (Vazyme or Takara) and 2 μL homemade T7 Transcription 5X buffer (400 mM HEPES-KOH pH 7.5, 120 mM MgCl$_2$, 10 mM spermidine and 200 mM DTT) at 37 °C for 4.5 h. After IVT, dsRNA was purified with LiCl precipitation and RNA concentration was quantified by agarose gel electrophoresis and nanodrop. dsRNA was injected into young adult hermaphrotides with concentration at 1000 ng/μL for *ref-2* and with concentration at 300 ng/μL for *vab-15* respectively. Strain *thuSi553*[P*clec-252*::neonGreen::H2B::*let-858* 3′UTR];*stIs10631*[*ref-2*a::H1::mCherry] were used for G2 phenotyping and *juIs76*[P*unc-25*::GFP];*stIs10631*[*ref-2*a::H1::mCherry] for W phenotyping (Supplementary Data 7).

## 2D fluorescence microscopy

For live imaging, worms were placed on a 2% agarose pad and immobilized using levamisole (2.5 mg/mL). Image was captured using a Zeiss Imager A2 epifluorescence microscope or confocal microscope either with Zeiss 780 or Leica sp8.

## Embryo lineage tracing

Embryo preparation and mounting were performed according to a previously described procedure[76]. 4D imaging were performed under a ×60 objective (PLAPON 60XO) at 20 °C ambient temperature using a spinning-disk confocal microscope (Revolution XD). Each embryo was taken for 30 focal ($z$) planes with 1-μm spacing per each scan, 75 s interval per each timepoint. The *unc-39* (*gk798*) homozygous mutants were identified by without expression of *qIs51*[P*myo-2*::GFP] marker. Images were processed with the StarryNite program[77,78] for automated cell identification and tracing to reconstruct embryonic cell lineages, followed by multiple rounds of manual inspection and editing using the AceTree program[79].

## Statistics and reproducibility

In all data presented in this paper, box plots illustrate statistical distributions as follows: the upper and lower hinges signify the first and third quartiles, respectively. Whiskers extend to the smallest and largest values, provided they are no more than 1.5 times the interquartile range from the hinges; points outside this range are identified as outliers and plotted individually. The median is indicated by a horizontal line within the box. No statistical method was used to

predetermine sample size. No data were excluded from the analyses. The experiments were not randomized, and investigators were not blinded to allocation during experiments and outcome assessment.

## Reporting summary

Further information on research design is available in the Nature Portfolio Reporting Summary linked to this article.

## Data availability

The worm image stacks generated in this study have been deposited in Zenodo (https://doi.org/10.5281/zenodo.7628038[80]). The datasets utilized for training the digital worm templates, as well as the trained templates employed in this study, are available on GitHub and have been archived with Zenodo for long-term accessibility (https://doi.org/10.5281/zenodo.8399308[81]). Source data are provided with this paper.

## Code availability

The software VANO can display and query the 3D digital template of the annotated 100 training worm stacks, two sets of 100 testing ones and point-cloud of the five templates used by our RAPCAT for cell annotation. The RAPCAT uses MATLAB (R2019a) and is available at Github with Zenodo DOI (https://doi.org/10.5281/zenodo.8399308[81]). The CellExplorer image analysis pipeline for computationally straightening nematode image stacks and the integrated function expressionAnalyzer for nematode gene expression analysis are available on GitHub with Zenodo DOI at (https://doi.org/10.5281/zenodo.8399277[12,82]).

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

## Acknowledgements

The authors thank Qianqian Feng from the Center of Biomedical Analysis, Tsinghua University and Xianyong Sheng from the Equipment Platform, Capital Normal University for confocal imaging. We are grateful to Dr. Hanchuan Peng from the Allen Institute for Brain Science and Dr. Start Kim for helpful discussions and comments on processing image data and biological results, respectively. The *Caenorhabditis*

Genetics Center and the National BioResource Project of Japan provided transgenic worms. This work was supported by the National Natural Science Foundation of China grants 32070639, 31871472, and 61871411, awarded to X.L., and grant 32200486, awarded to D.Z.; Ministry of Science and Technology of China Grant 20131970194 and the University Synergy Innovation Program of Anhui Province [GXXT-2019-008 and GXXT-2021-001] awarded to L.Q.

## Author contributions

X.L. conceived the study. Yb.L., S.C., W.L., D.Z., Y.G., S.H., H.L. and Yy.L. performed the experiments. Yb.L., L.Q. and X.L. constructed image computation. X.L. wrote the manuscript.

## Competing interests

The authors declare no competing interests.
