## [Peer Review File · Nature Communications]

A full-body transcription factor expression atlas with completely resolved cell identities in *C. elegans*REVIEWER COMMENTS

Reviewer #1 (Remarks to the Author):

Li et al. describe a tool, RAPCAT, to automatically identify cell position. The authors then go on to generate an atlas of expression of hundreds of *C.elegans* transcription factors. Analyzing these expression patterns, they describe a number of vignettes of potential biological significance, such as the identification of potentially quiescent stage of the HSN neurons. The authors also make a number of conclusions in regard to the relationship of TF expression and lineage.

The paper has great potential for three reasons:

(1) the RAPCAT tool

(2) the expression patterns serve as great hypothesis building tools for future functional analysis

(3) the reporter transgenes serve as value cell fate tools

This being said, there are substantial potential problems with the paper that the authors need to address before I can make a suggestion on suitability for publication in Nature Comm.

Problem #1: the cell ID tool RAPCAT has not been properly validated for whether it can indeed reliably identify cells, particularly in the densely packed brain of the worm. This is absolutely essential. The authors need to do such validation with reporters whose expression has been unambiguously and correctly identified. One set of reporters that comes to mind are the homeodomain TF reporter alleles published by Reilly et al. in 2020. Many dozens show very selective expression in the nervous system. These expression pattern have been identified in L4/young adults, but there are no differences in expression at the L1 stage. The author need to see whether RAPCAP can properly identify these patterns. The authors shall not limit this to a single reporter, but to a substantial number of reporters that in aggregate cover most of the nervous system.

Problem #2: While the reporter collection established by the authors is potentially useful, it is a very significant shortcoming that the vast majority of reporters used are mere promoter fusions that very often do not capture the full expression of the respective genes. This is a fundamental problem in the interpretation of the expression patterns. The authors are in a quite unique position to address such discrepancies, by again making use of the homeobox reporter alleles mentioned above, and comparing their published expression patterns with the many homeobox promoter gene fusions that they use in the paper. The comparison will give a sense of how accurate a reflection of real expression patterns these reporters are. Again, such comparison have to be done with proper sample sizes, e.g. the entire homeobox family down by Reilly et al., with the promoter-based expression patterns of homeobox genes by the present authors. One or two genes are not enough. Same random sampling of patterns that I did manually make me worry about quite substantial disparities, but this may be biased.

If there are indeed substantial discrepancy between the expression patterns, the author will have to dampen a large number of their conclusions.

Two other comments:

- For the reporter data, the authors should not just give primer sequence, but coordinates relative to start codon of the gene so that it is easier to grasp how much genomic region is covered by the reporter.

- one of the greatest values of this paper is the reporter resource. I expect that according to

journal policy, these strains are deposited in a strain repository (CGC).

Reviewer #2 (Remarks to the Author):

Li et al reports a TF expression atlas in *C. elegans* with single-cell resolution, covering 612 TFs whose reporter expression is mapped to each of the 558 cells in the newly hatched L1 larvae. It is an impressive body of work with unique values complementing the more commonly used scRNA atlases. The authors offered a systemic analysis of the expression patterns in terms of terminal differentiation, lineage history and body axes, and identified an interesting case of developmental quiescence. The study also established a computational approach to reliably identify individual cells based on nuclear positions, which is still difficult to do in *C. elegans*.

The text is well written and easy to follow. However, the figures are hard to read. The labels are too small to read. On the other hand, some figures (eg, 4 and 5) show excessive images -- it would be more effective to show the quantification with a representative image while moving the entire montage to a sup figure. This way it might also be possible to merge figures 4 and 5 into one, and split Fig 3 to two for the real estate it needs.

Most of the comments are on the computational method of template matching, which lacks crucial details:

The presentation of the method in the main text is a bit unclear about how the full set of examples is used in scoring, this could and should be included in overview in the main text.

The choice of primary alignment template for the atlas appears to have an, at first glance, unusually large influence on the quality of results. It would be helpful to give some intuitions why this is, and what characteristics the other 4 sub-optimal templates chosen have (from fig S2A they appear to have perhaps been chosen for their unusually good -worst case- performance, though the caption suggests they are the runners up in fig S2b, this should be clarified a bit more). Regardless of method what does the choice of 5 and templates represent? Major modes of position? Could a different metric for co-alignment of the 100 examples (rather than least squares fit to one chosen example) create a single reference atlas with different/better properties. What are the intuitions that argue for this approach rather than doing RPM 100 times and voting over these answers.

It's valuable that the authors quantified the error rate of the overall annotation pipeline including screening and correction for low confidence matches. Can they provide some insight about what influence the residual error of around 3% might have on results, and what minimal accuracy is needed?

Is the 95% cited for raw alignment sufficient?

How long does exhaustive curation take and how long does curating low confidence matches take?

Reviewer #3 (Remarks to the Author):

In this herculean work, Li and co-authors have generated an extensive collection of promoter-driven fluorescent reporter lines of transcription factors (TFs) and presented an expression atlas of TF reporters that cover 65% of all predicted TFs in every lineage-resolved single cell of the L1 *C. elegans* larvae. Although scRNA-seq approaches have been previously used to determine high-resolution transcriptomes during *C. elegans* embryonic (e.g., Packer et al. Science 2019) and post-embryonic development (e.g., Cao et al. Science 2017 and Taylor et al. Cell 2021), only a small portion of the transcriptomes can be unequivocally linked to a specific cell. This uncertainty undermines the analysis of cell type-cell state relationships and state heterogeneity among similar/identical cell types. In contrast, this study takes advantage of the invariant cell lineage and stereotypic cell positioning in *C. elegans* development, using an imaging-based approach that allows for the quantification of the expression of a gene of interest in every annotated single cell and the synthesis of individual expression patterns into an atlas. The key advantage of this dataset is the clarity of cell annotation and complete cell coverage. Given the functional importance and expression informativeness of TFs, this dataset provides an opportunity to define the molecular/regulatory state of every cell at a developmental stage. Although Imaging-based single-cell analysis of TF expression at comparable resolution and scale has been conducted in *C. elegans* embryos (Murray et al. Genome Research 2012 and Ma et al. Nature Methods 2021), this study stands out as the first to generate a comprehensive TF expression atlas at the L1 stage, with a greater number of TFs and higher cell coverage.

Using the atlas, the authors tackle an important yet unresolved question regarding the complex relationships between cell types and states. They grouped cells based on their TF expression (molecular state) and compared them to classic classifications based on the morphology and function of cells, reporting both consistencies and discrepancies between the two classifications. The authors described three cases to demonstrate the usefulness of the atlas in uncovering new cell states and state-type relationships. The first case describes a previously unrecognized quiescent state in newborn HSN neurons, characterized by gene repression. The second case concerns the state heterogeneity between P and G2/W neuroblasts, in which the authors characterized the regulatory differences between the otherwise similar cell types. Finally, the authors analyzed the incomplete convergence of cell states between cells originating from unrelated cell lineages but differentiating into identical/similar cell types, a phenomenon previously described during embryogenesis. Here, the authors verified and extended previous findings, showing that the lineage-dependent differences in cell states (TF expression) persist at the L1 stage when the terminal differentiation of many cells is completed. Furthermore, they focused on muscle cells and identified that earlier expression TFs (*unc-39* and *hnd-1*) are responsible for lineage-dependent state heterogeneity through asymmetric functions in different lineages.

This study presents a valuable functional genomic resource for studying multicellular development. The over 900 reporter lines (including ~700 transgenic lines generated in this study) and associated cellular expression patterns will be valuable genomic resources for investigating TFs, cell fates, and cellular states. The experiments are generally well executed, and the results appear to be of high quality. The comparisons of cell states defined by TF expression to cell types are useful, and the case studies are interesting. However, the presentation of the results in many sections is not well organized and presented, making some parts of the manuscript hard to follow and unclear the key message the authors want to deliver. Additionally, the quality assessments of the expression data are somewhat insufficient, and the authors tend to overlook the caveats of using fluorescent

reports to indicate gene transcription, which should be explicitly stated and discussed. Finally, while analysis of cellular states using the TF atlas is definitely a good direction, the authors missed the rich opportunity of using this atlas to prioritize and characterize new regulators of cell types. Including this analysis could further enhance the impact of this study. To improve the manuscript, I suggest the following revisions.

Major comments:

1. RAPCAT methods. The authors provided a detailed description of the development of the RAPCAT algorithm and its ability to automatically assign cell identities based on stereotypic relative cell positions using training datasets. However, the accuracy of cell segmentation is equally important as well, which is not mentioned. Therefore, the authors should provide more information on how they dealt with the errors in nuclei segmentation, which inevitably affects the robustness of cell matching.

2. Quality assessment of expression atlas. To increase the utility of the atlas, it is critical to provide a thorough quality evaluation of the data to give readers a general idea of its quality. Although the authors show that the technical reproducibility of expression patterns is reasonably high ($R > 0.81$ between replicates), this result does not necessarily validate the correctness of the expression. Indeed, the correlation between different reporters of the same TF is not high (Figure 2B), raising concerns about to what extent the reporter expression recapitulates the endogenous expression. Since endogenous fluorescent fusion lines are available for many of the TFs analyzed in this study, the authors should perform a small-scale comparative analysis to determine the consistency. In addition, the authors should compare their expression pattern to benchmark genes whose expression and function are relatively well-documented. Although the single-cell expression at the L1 stage is not available for most TFs, tissue-level analysis is feasible.

3. Caveats of reporter-based expression profiling. The authors should explicitly mention and discuss the limitations of the fluorescent reporter-based strategy. Although this approach is a reasonable choice, it has several caveats that compromise the accuracy of the expression data. These include but are not limited to (1) the inability to fully recapitulate endogenous expression due to the relatively short promoter sequence included in the reporter and missing critical regulatory elements, (2) false negatives due to low detection sensitivity of imaging, (3) the potential influence of the integration site on expression, and (4) some of the expression signals are not from the subject cell but inherited from earlier cells through the cell lineage due to long half-life of the fluorescent protein. Importantly, some of the caveats mentioned above may also affect the reliability of TF-based cell clustering results. For example, the long half-life of fluorescent protein would cause an overestimation of the influence of cell lineage on cellular states, which should be taken into account when interpreting the results (see below).

4. Cell clustering by TF expression. The key finding of this study is presented in Figure 3, but the results are not well-organized, making it difficult to follow. The use of many different colors in the figure makes it impossible to understand the exact relationships between different cell classifications. Additionally, the presentation of the different cell classifications in Table S4 is unclear. In particular, organizing the classifications into a matrix is highly unintuitive. A more effective way to present the information would be to show each cell and then different clustering or classifications as numbers or text.

5. Gene-level analysis. The cellular expression patterns of individual TFs are informative in revealing regulators of specific cell fates, types, or states. However, the authors did not

perform any analysis to characterize TFs with interesting yet uncharacterized expression patterns. At the very least, the authors should identify and discuss individual TFs exhibiting cell-type or cell-state-specific expression patterns. This analysis is critical because it helps to demonstrate the value of high-resolution expression analysis to identify novel expression enrichment/functions of TFs. Furthermore, it is also an integral part of quality control to show how many known patterns are recapitulated (see Major comment 2).

6. Quiescent HSN state. TF expression status in the sister cell of HSNs (PHBs) should be included as an internal control to demonstrate gene repression in the quiescent HSN state. As the authors use the fluorescent signal to monitor gene transcription, protein turnover is likely involved in addition to repressing gene expression, and this possibility should be discussed. Moreover, if possible, the authors should use more endogenously-tagging reporters to verify the proposed gene repression in HSN neurons. Furthermore, since HSNs are hermaphrodite-specific neurons (the equivalent cells undergo programmed cell death in males), it is worth discussing whether apoptosis-related pathways play a role in gene repression.

7. P and G2/W states. This section is difficult to follow, and it is unclear what the authors intended to highlight. If I understand correctly, the authors identified P and G2/W as two subtypes of neuroblasts. They then conducted follow-up experiments and characterized the shared and distinct molecular regulation of the two states by several known regulators (ref-2, vab-15, and tlp-1). At first glance, the authors seem to highlight the molecular distinction of the subtypes. However, the authors concluded the findings as "... revealed a TF battery (ref-2, vab-15, and tlp-1) shared between the TF-based G2/W and P neuroblast subtypes.". In addition, it is unclear what hypothesis the authors are trying to form in this long sentence: "So we speculated that two neuroblast subtypes specified by similar pro-neural TF batteries would have smaller inter-subtype difference in TF expression normalized by their cross-sister clade difference than that between subtypes specified by different TF batteries (Figure S5D)." This section should be significantly revised to make it clear what the hypothesis is and how each finding supports it.

8. Lineage-dependent heterogeneity. As mentioned in Major Comment 3, the long half-life of fluorescent proteins will make early expressed proteins persist in descendants through the cell lineage even though the endogenous transcripts or proteins no longer exist. As a result, the lineage effect may be overestimated since the propagation of protein is through mitotic divisions in each lineage. To address this concern, the authors should validate their findings by focusing on reporters that are expressed since the L1 stage, thereby avoiding this caveat.

Minor Comments:

Line 83: "Moreover, a significant source of this intra-type molecular heterogeneity was convergence of different lineages into the same cell fate, i.e., multiple cell lineages could produce cells of the same phenotype and function." – A complete convergence should not produce heterogeneity.

Line 207-211: Is the integration site the only difference between these strains?

Line 218: The correlation coefficient should be provided instead of being displayed as a color gradient.

Line 225: Figure 3E-G should be Figure 2E-G, also Figure 2H is not cited before Figure 3

Line 257: sister-clade cells, To avoid confusion with sister cells, I would suggest using a term like "neighbor-clade cells" or something similar

Lin 1190-1192: "The mean expression profile across all image stacks of a strain was used as the strain expression profile. The mean expression profile across all strains of a TF was used as expression profile of the TF." – While it is reasonable to average expression across experimental replicates, it is uncertain whether it is valid to average expression between different reporters for the same transcription factor, especially if the expression levels differ significantly

Figure 3A: Rather than organizing the hierarchical clustering into a circular-shaped plot, a classic horizontal view would be more intuitive and effective. Additionally, the current plot does not allow for easy visualization of which cells belong to the same clade

Line 324: in the paragraph, Figure 6 should be Figure 5.

Line 370: "It is well-established that 97 of these 159 cell pairs are symmetric in cell lineage, while the other 62 pairs are converged cell pairs, whose left and right cell mates have an asymmetric cell lineage history." –To clarify the meaning of "symmetric" and "asymmetric," an example should be provided.

Reviewer #1 (Remarks to the Author):

Li et al. describe a tool, RAPCAT, to automatically identify cell position. The
authors then go on to generate an atlas of expression of hundreds of C.elegans
transcription factors. Analyzing these expression patterns, they describe a
number of vignettes of potential biological significance, such as the
identification of potentially quiescent stage of the HSN neurons. The authors
also make a number of conclusions in regard to the relationship of TF
expression and lineage.

The paper has great potential for three reasons:

(1) the RAPCAT tool

(2) the expression patterns serve as great hypothesis building tools for future
functional analysis

(3) the reporter transgenes serve as value cell fate tools

This being said, there are substantial potential problems with the paper that
the authors need to address before I can make a suggestion on suitability for
publication in Nature Comm.

Problem #1: the cell ID tool RAPCAT has not been properly validated for
whether it can indeed reliably identify cells, particularly in the densely packed
brain of the worm. This is absolutely essential. The authors need to do such
validation with reporters whose expression has been unambiguously and
correctly identified. One set of reporters that comes to mind are the
homeodomain TF reporter alleles published by Reilly et al. in 2020. Many
dozens show very selective expression in the nervous system. These expression
pattern have been identified in L4/young adults, but there are no differences in
expression at the L1 stage. The author need to see whether RAPCAP can
properly identify these patterns. The authors shall not limit this to a single
reporter, but to a substantial number of reporters that in aggregate cover most
of the nervous system.

We agree that homeodomain TF reporter alleles published by Reilly et al. in 2020 could
serve as reliable and well-defined positive controls to validate the annotation accuracy
of RAPCAT since their neuronal expression patterns were determined through well-
established markers. As recommended, we obtained 11 of these TF reporters from
colleagues in the worm research community. Among the 222 neurons already present
in L1 larvae, these reporters were expressed in 1 – 108 neurons of young adults
(Supplementary Figure 4E). We generated image stacks of the 11 reporter strains and
annotated them using RAPCAT without the manual EPC step (*i.e.*, the stacks were
annotated automatically). One stack, capturing *hmbx-1* expression, did not pass the
0.995 WAC threshold and was therefore excluded from the validation of automated
RAPCAT annotation accuracy. For the remaining 10 reporters, RAPCAT annotation
indicated that a TF was simultaneously expressed in at least 1 and as many as 80
neurons. Among the 222 total neurons, 140 showed detectable GFP expression by at
least one reporter at the L1 stage in our 10-TF profile. Among the 140 GFP-expressing

neurons in our results, 128 were also positive for expression of all corresponding
reporters in young adults (Supplementary Figure 4E), suggesting 91.4% consistency
between the automated neuron annotation by RAPCAT and marker-based identity
assignment by NeuroPAL.

Transcriptomic changes have been previously reported in the post-mitotic nervous
system during larval development (Sun and Hobert, 2021). It is thus reasonable to
speculate that some discrepancies between RAPCAT annotation based on L1
expression profile and the adult profile reported by Reilly may be due to temporal
changes in gene expression rather than error in RAPCAT annotation. We conducted a
manual examination of the RAPCAT annotation outcomes for the 12 neurons that
exhibited detectable GFP expression exclusively in L1 larvae, without the same
expression observed in young adults. Based on our thorough inspection, it was
determined that seven out of these 12 neurons had received accurate cell identity
assignment through the RAPCAT annotation process. In other words, automatic
RAPCAT annotation errors were present in only five neurons. These included a tail
neuron PHB, a pharyngeal neuron I6, and three neurons (ASGL, AVDL, and AVHR)
located around the nerve ring (Supplementary Figure 4E). Collectively, the automatic
RAPCAT annotation method effectively and unambiguously identifies neurons in L1
larvae, achieving an accuracy ranging from 91.4% to 96.4%, encompassing even the
densely packed neurons in the brain.

We have added these new data to Lines 165 - 169 of the revised manuscript and Lines
51 - 88 of the revised Supplementary Information.

Problem #2: While the reporter collection established by the authors is
potentially useful, it is a very significant shortcoming that the vast majority of
reporters used are mere promoter fusions that very often do not capture the
full expression of the respective genes. This is a fundamental problem in the
interpretation of the expression patterns. The authors are in a quite unique
position to address such discrepancies, by again making use of the homeobox
reporter alleles mentioned above, and comparing their published expression
patterns with the many homeobox promoter gene fusions that they use in the
paper. The comparison will give a sense of how accurate a reflection of real
expression patterns these reporters are. Again, such comparison have to be
done with proper sample sizes, e.g. the entire homeobox family down by Reilly
et al., with the promoter-based expression patterns of homeobox genes by the
present authors. One or two genes are not enough. Same random sampling of
patterns that I did manually make me worry about quite substantial disparities,
but this may be biased.

We appreciate this highly constructive suggestion. It is well-established that promoter-
fusion reporters may not always fully recapitulate the expression patterns of their
corresponding genes because the necessary regulatory elements are located outside
the upstream sequence used in the construct, such as those in introns (Bradnam *et al.*,

2008; Fuxman Bass *et al.*, 2014) or distal enhancers in long intergenic sequence
(Okkema and Krause, 2005). Similarly, the transcription of downstream genes in an
operon is usually controlled by a promoter outside of its direct upstream intergenic
sequence (Okkema and Krause, 2005), and is consequently excluded from screens of
the immediate upstream region, such as our promoter fusions. We therefore tested the
extent of their influence by comparing the profiles of 84 TFs shared between our L1
profile and the adult profiles reported by Reilly and colleagues (2020). This set
included 58 promoter-fusion reporters, 22 fosmid-based reporters, and 4 knock-in
reporters (Supplementary Data 1), with fosmid and knock-in reporters serving as gold
standards because they contained the full genomic context of the respective genes.

Target genes were then classified into four types, *i.e.*, upstream intergenic
sequence >7kb, upstream intergenic sequence <1kb, genes with >1kb intron following
the start codon, and all other genes. The cloned promoter regions of genes categorized
in the fourth group were presumed to contain the necessary and sufficient regulatory
elements for driving their expression, termed as high context promoter reporters.
Substantiating this proposition, the L1 profiles of these high context promoter
reporters displayed strong correlations (median of ROCAUC = 0.92) with
corresponding adult homeobox TF profiles as reported by Reilly and colleagues. This
correlation level resembled the relationship observed between the gold standard
fosmid/knock-in reporters and the adult profiles (median of ROCAUC = 0.90) (Figure
2C). In our study, these closely correlated promoter reporters constituted a total of 296
TFs. In contrast, promoter fusions belonging to the first three categories, referred to as
low context promoter reporters, displayed weak correlations with adult stage profiles
by Reilly *et al.* (Figure 2C). These low-context promoter reporters encompassed 263
TFs (Supplementary Data 1).

These data have been added to lines 251 - 277 of the revised manuscript.

If there are indeed substantial discrepancy between the expression patterns,
the author will have to dampen a large number of their conclusions.

We are very grateful for the reviewer's advice.

As explained in the initial section of Question #2, we categorized the reporters of the
620 profiled TFs into three distinct types based on the extent to which their reporter
constructs encompassed regulatory elements (Figure 2C). These categorized comprised
61 fosmid/knock-in reporters, 296 high-context promoter reporters, and 263 low-
context promoter reporters (Supplementary Data 1). Upon conducting further analysis
and comparing the results with the adult stage homeobox TF profiles reported by Reilly
and colleagues, it became evident that both fosmid/knock-in and high-context promoter
reporters effectively captured the full expression of their respective genes. In contrast,
low-context promoter reporters demonstrated relatively weaker correlations with the
published adult stage profiles.

We conducted an analysis to measure the alignment between traditional cell
classifications and TF-driven cell clustering, aiming to quantify the impact of
discrepancies between reporter expression and endogenous gene expression on our
interpretations of biological function. It is noteworthy that all biological implications
explored within this study were based on discrepancies between phenotype-based
classification and molecular clustering. Our findings revealed that out of the total 119
multi-cellular phenotypic cell classes, fourteen classes were distributed across multiple
clades in the dendrogram (Figure 3 and Supplementary Data 4). This distribution
pointed to disparities between phenotype-based classification and molecular clustering,
specifically relying on the profiles of all 620 TFs that were examined. Upon excluding
the 263 low-context promoter reporters from our profiles, additional six cell classes
exhibited discordance between phenotype-based classification and TF-based clustering
(Supplementary Figure 6L). This suggests that low-context promoter reporters could
potentially offer insights in the gene regulation program sensed by specific DNA
sequences, even if they did not entirely encompass the expression patterns of their
corresponding endogenous genes.

Notably, when we excluded reporters that were likely comprehensive in capturing full
expression of their respective genes (specifically, all 61 fosmid/knock-in reporters and
a random selection of 202 high-context promoter reporters), it is intriguing that the
count of cell classes exhibiting discordance between phenotype-based classification
and TF-based clustering remained comparable to that observed when profiles excluding
the 263 low-context promoter reporters were considered (18 vs. 20) (Supplementary
Figure 6L). Moreover, we found molecular clustering of the large set of 263 low-context
reporters could recapitulate well-established phenotype-based cell type classifications
to strikingly higher extent than a smaller set of 61 fosmid/knock-in reporters
(Supplementary Figure 6K). This finding led us to conclude that the number of profiled
genes affected TF-based cell type identification more strongly than full genomic
context. While we have included our findings of low- and high-context reporter profiles
in the revised manuscript, we respectfully maintain the high number of reporter profiles
for reliable TF-based cell type annotation.

These data have been added to Lines 91 – 99 and Lines 119 – 147 of the revised
Supplementary Information.

Two other comments:

- For the reporter data, the authors should not just give primer sequence, but
coordinates relative to start codon of the gene so that it is easier to grasp how
much genomic region is covered by the reporter.

**Supplementary Data 1 has been updated to include the suggested information.**

- one of the greatest values of this paper is the reporter resource. I expect that
according to journal policy, these strains are deposited in a strain repository
(CGC).

We completely agree. The reporter strains will indeed be available to the worm
research community, and the set may be further expanded as the tool is refined.

We would like to thank Reviewer #1 for their time and careful reading of our
paper, and for their highly constructive remarks, which have helped us to
bolster the rigor of our analyses and ultimately strengthen our conclusions.

Reviewer #2 (Remarks to the Author):

Li et al reports a TF expression atlas in *C. elegans* with single-cell resolution,
covering 612 TFs whose reporter expression is mapped to each of the 558 cells
in the newly hatched L1 larvae. It is an impressive body of work with unique
values complementing the more commonly used scRNA atlases. The authors
offered a systemic analysis of the expression patterns in terms of terminal
differentiation, lineage history and body axes, and identified an interesting case
of developmental quiescence. The study also established a computational
approach to reliably identify individual cells based on nuclear positions, which
is still difficult to do in *C. elegans*.

The text is well written and easy to follow. However, the figures are hard to
read. The labels are too small to read. On the other hand, some figures (eg, 4
and 5) show excessive images -- it would be more effective to show the
quantification with a representative image while moving the entire montage to
a sup figure. This way it might also be possible to merge figures 4 and 5 into
one, and split Fig 3 to two for the real estate it needs.

As suggested, Figure 3 was split into Figures 3 and 4 in the revised manuscript, which
has allowed us to enlarge the labels of cell clusters. Original Figure 4 was moved to
Supplementary Figures 5A-5I and HSN expression is shown as dot plots in Figure 4I.

Most of the comments are on the computational method of template matching,
which lacks crucial details:

The presentation of the method in the main text is a bit unclear about how the
full set of examples is used in scoring, this could and should be included in
overview in the main text.

We are grateful for this constructive suggestion. How the full set of 100 training image
stacks was used to build 100 templates is summarized in the original caption of
Supplementary Figure 2. This content has been moved to the main text with expanded
details.

Specifically, one training worm was selected as the initial target and all straightened

worm stacks were registered to the target using global affine transformation, resulting
in a template L1 cell organization (Supplementary Figures 1D-1G). This process was
repeated 100 times, with each training stack used once as the initial target for template
generation, resulting in 100 digital templates of cell organization for L1 worms.

These data have been added to Lines 92 - 97 of the revised manuscript.

The choice of primary alignment template for the atlas appears to have an, at
first glance, unusually large influence on the quality of results. It would be
helpful to give some intuitions why this is, and what characteristics the other 4
sub-optimal templates chosen have

We appreciate this valuable question. We selected multiple templates instead of a single
reference atlas because the distribution of cell nuclei does not have a strict stereotype
(*i.e.*, can vary among individual worms). For instance, variations in developmental
program have been well-documented to result in slight variations in the exact location
of cell nuclei (Sulston and Horvitz, 1977; Sulston *et. al.*, 1983; Yemini *et. al.*, 2021).
Furthermore, cell identity can determine potential variability of developmental program
for individual cells. For example, a ring of four nuclei is invariably found surrounding
the lumen in the anterior-most region of the intestine (Sulston and Horvitz, 1977). By
contrast, the arrangement of neurons anterior to the intestine shows relatively high
variability among newly hatched L1 larvae (Sulston *et. al.*, 1983). In addition, worms
adopt different curvatures at the time of fixation for confocal imaging. As a result, their
3D image stacks require computational straightening so that they can be annotated by
a canonical coordinate system. The straightening process could potentially deform
relative cell locations (Peng *et. al.*, 2008). In short, several factors, such as cell identity-
related differences in developmental program, and artifacts of the image straightening
process for curved worm bodies can collectively contribute slight differences in cell
positions among stereotype templates.

In practice, the 5-template-based annotation showed higher accuracy and more robust
annotations compared to the optimal-template-based results in tests of 100 image stacks
(Figure 1F), thus supporting our intuition that multiple templates can better capture
inter-prototype variation in the positions of cell nuclei.

These data have been added to Lines 24 – 44 in the revised Supplementary Information.

(from fig S2A they appear to have perhaps been chosen for their unusually
good -worst case- performance, though the caption suggests they are the
runners up in fig S2b, this should be clarified a bit more).

We thank the reviewer for pointing out this unclear language. These templates were
indeed selected based on their strong performance in annotating stacks that were ‘worst
case’ examples for the optimal template. To avoid confusion, we have revised the

caption of Supplementary Figure 2B as follows: These templates that provided the best
performance in annotating worm stacks that could not be readily annotated by the
optimal template (blue) were then adopted as the alternative templates. (See
Supplementary Figure 2B caption, Lines 243 – 246 in the revised Supplementary
Information)

Regardless of method what does the choice of 5 and templates represent?
Major modes of position?

As explained above, we hypothesized that computationally straightened image stacks
could have slightly different stereotypical cell positions and that 5 templates could
accommodate the range of variability in nuclei positions resulting from developmental
differences or the fixation process. It is important to highlight that the disparity in mean
and variance of cell positions among these five templates was exceedingly simple
(Supplementary Figures 2E and S2F). Nonetheless, this subtle difference yielded
distinct matching scores (Supplementary Figures 2G and 2H), underscoring the
necessity for separate templates to effectively assign cell identities.

These data have been added to Lines 44 – 49 in the revised Supplementary Information.

Could a different metric for co-alignment of the 100 examples (rather than least
squares fit to one chosen example) create a single reference atlas with
different/better properties.

We thank the reviewer for this thoughtful comment. Least-squares matching is a well-
developed image registration method (Pattern recognition 32: 1783, 1999; International
Archives of Photogrammetry, Remote Sensing and Spatial Information Sciences 38:86,
2010; Automatica 73: 155, 2016, and Journal of WSCG 25:21, 2017). Moreover, global
affine transformation using least squares has been established as a reliable method for
accurate identity assignment of trunk and tail cells in worm image stacks (Nature
Methods 6:667, 2009). We therefore decided to build upon the least-squares matching
method in the current study.

However, we agree with the reviewer's suggestion that a single, consensus reference
atlas with minimal position bias towards a specific worm image stack could be valuable
for cell annotation. We therefore followed a 3-step process in which each worm training
stack was first aligned to the optimal template (*i.e.*, template #2) by global affine
transformation. Subsequently, we computed the average deformation field from these
transformed stacks and inverted it. This inverted parameter was then utilized to deform
the optimal template, resulting in a new template. Finally, step one was reiterated and
the new template replaced the optimal template. This iterative process continued until
the template converged into a stable template, which was designated as the consensus
template.

Unexpectedly, the consensus template showed lower accuracy than the 5-template set

(95.3% vs. 95.9%) in tests of 100 image stacks, and fewer worm stacks passed WAC
threshold (85 vs. 94) (Supplementary Figure 2C). Based on these results, we retained
the 5-template method rather than the consensus template in further analyses.

These data have been added to Lines 104 – 114 and 147 – 150 in the revised manuscript.

What are the intuitions that argue for this approach rather than doing RPM 100
316 times and voting over these answers.

We appreciate this valuable question. We found that annotations based on 100 templates
did not result in higher accuracy cell identity assignment than annotation with the 5-
template set (Supplementary Figure 2D), but was markedly more time-consuming due
to the greater computational burden, leading us to abandon this strategy.

These data have been added to Lines 183 – 190 of the revised manuscript.

It's valuable that the authors quantified the error rate of the overall annotation
pipeline including screening and correction for low confidence matches. Can
they provide some insight about what influence the residual error of around 3%
might have on results, and what minimal accuracy is needed?

We analyzed agreement between classic cell classifications and TF-based cell clustering
to quantify the impact of annotation error on our inference of biological function since
all biological implications examined in this study were based on discrepancies between
phenotype-based classification and molecular clustering.

We found that fourteen of 119 multi-cellular phenotypic cell classes were distributed
across multiple clades in the TF-based dendrogram (Figure 3 and Supplementary data
4), indicating discordance between phenotype-based classification and molecular
clustering. The accuracy of annotations for cells in these discordant phenotypic classes
were significantly lower than that of concordant cell classes (Supplementary Figure 6J),
suggesting the influence of annotation error on cell clustering

These data have been added to Lines 93 – 103 in the revised Supplementary Information.

Is the 95% cited for raw alignment sufficient?

In order to measure the impact of annotation errors on cell clustering, we generated TF
profiles using five RAPCAT modes with annotation accuracy rates ranging from 88%
to 97% (Figure 1H). Subsequently, we clustered cells using these profiles. The effect of
annotation accuracy on the agreement between phenotype-based classification and
molecular clustering was marginal when annotation accuracy ranged from 93% to 97%
(Supplementary Figure 6K).

These data have been added to Lines 103 – 109 in the revised Supplementary
Information.

How long does exhaustive curation take and how long does curating low
confidence matches take?

For an image stack passing the 0.995 WAC threshold, EPC curation of low confidence
matches typically requires about a quarter of an hour for a well-trained annotator. For
image stacks that do not pass the WAC threshold, manual annotation of all 558 cells is
necessary, often requiring more than two hours.

These data have been added to Lines 197 – 199 of the revised manuscript.

We would like to thank the reviewer for his careful attention to detail and insightful
questions that have given us much to consider and helped strengthen our conclusions.

Reviewer #3 (Remarks to the Author):

In this herculean work, Li and co-authors have generated an extensive
collection of promoter-driven fluorescent reporter lines of transcription factors
(TFs) and presented an expression atlas of TF reporters that cover 65% of all
predicted TFs in every lineage-resolved single cell of the L1 *C. elegans* larvae.
Although scRNA-seq approaches have been previously used to determine
high-resolution transcriptomes during *C. elegans* embryonic (e.g., Packer et al.
*Science* 2019) and post-embryonic development (e.g., Cao et al. *Science*
2017 and Taylor et al. *Cell* 2021), only a small portion of the transcriptomes
can be unequivocally linked to a specific cell. This uncertainty undermines the
analysis of cell type-cell state relationships and state heterogeneity among
similar/identical cell types. In contrast, this study takes advantage of the
invariant cell lineage and stereotypic cell positioning in *C. elegans*
development, using an imaging-based approach that allows for the
quantification of the expression of a gene of interest in every annotated single
cell and the synthesis of individual expression patterns into an atlas. The key
advantage of this dataset is the clarity of cell annotation and complete cell
coverage. Given the functional importance and expression informativeness of
TFs, this dataset provides an opportunity to define the molecular/regulatory
state of every cell at a developmental stage. Although Imaging-based single-
cell analysis of TF expression at comparable resolution and scale has been
conducted in *C. elegans* embryos (Murray et al. *Genome Research* 2012 and
391 Ma et al. *Nature Methods* 2021), this study stands out as the first to generate
a comprehensive TF expression atlas at the L1 stage, with a greater number
of TFs and higher cell coverage.

Using the atlas, the authors tackle an important yet unresolved question
regarding the complex relationships between cell types and states. They

grouped cells based on their TF expression (molecular state) and compared
them to classic classifications based on the morphology and function of cells,
reporting both consistencies and discrepancies between the two
classifications. The authors described three cases to demonstrate the
usefulness of the atlas in uncovering new cell states and state-type
relationships. The first case describes a previously unrecognized quiescent
state in newborn HSN neurons, characterized by gene repression. The second
case concerns the state heterogeneity between P and G2/W neuroblasts, in
which the authors characterized the regulatory differences between the
otherwise similar cell types. Finally, the authors analyzed the incomplete
convergence of cell states between cells originating from unrelated cell
lineages but differentiating into identical/similar cell types, a phenomenon
previously described during embryogenesis. Here, the authors verified and
extended previous findings, showing that the lineage-dependent differences in
cell states (TF expression) persist at the L1 stage when the terminal
differentiation of many cells is completed. Furthermore, they focused on
muscle cells and identified that earlier expression TFs (*unc-39* and *hnd-1*) are
responsible for lineage-dependent state heterogeneity through asymmetric
functions in different lineages.

This study presents a valuable functional genomic resource for studying
multicellular development. The over 900 reporter lines (including ~700
transgenic lines generated in this study) and associated cellular expression
patterns will be valuable genomic resources for investigating TFs, cell fates,
and cellular states. The experiments are generally well executed, and the
results appear to be of high quality. The comparisons of cell states defined by
TF expression to cell types are useful, and the case studies are interesting.
However, the presentation of the results in many sections is not well
organized and presented, making some parts of the manuscript hard to follow
and unclear the key message the authors want to deliver. Additionally, the
quality assessments of the expression data are somewhat insufficient, and the
authors tend to overlook the caveats of using fluorescent reports to indicate
gene transcription, which should be explicitly stated and discussed. Finally,
while analysis of cellular states using the TF atlas is definitely a good
direction, the authors missed the rich opportunity of using this atlas to
prioritize and characterize new regulators of cell types. Including this analysis
could further enhance the impact of this study. To improve the manuscript, I
suggest the following revisions.

Major comments:

1. RAPCAT methods. The authors provided a detailed description of the
development of the RAPCAT algorithm and its ability to automatically assign
cell identities based on stereotypic relative cell positions using training
datasets. However, the accuracy of cell segmentation is equally important as

well, which is not mentioned. Therefore, the authors should provide more
information on how they dealt with the errors in nuclei segmentation, which
inevitably affects the robustness of cell matching.

We are grateful for the Reviewer's supportive comments regarding our study and for
their astute suggestions for improvement. We agree that correct segmentation is
indeed essential for accurate cell annotation. Since the segmentation software was
designed for the trunk and tail of L1 larvae (Long et. al. 2009), there are typically
numerous segmentation errors in the densely packed brain region of the worm that
require manual curation to resolve. Using VANO, a well-trained worm biologist may
require two hours to identify and curate all segmentation errors. An image stack
without segmentation errors has around 558 nuclear masks, depending on whether
there are more or less than 20 intestinal nuclei.

These data have been added to Lines 808 – 815 of the revised manuscript.

2. Quality assessment of expression atlas. To increase the utility of the atlas, it
is critical to provide a thorough quality evaluation of the data to give readers a
general idea of its quality. Although the authors show that the technical
reproducibility of expression patterns is reasonably high ($R > 0.81$ between
replicates), this result does not necessarily validate the correctness of the
expression. Indeed, the correlation between different reporters of the same TF
is not high (Figure 2B), raising concerns about to what extent the reporter
expression recapitulates the endogenous expression. Since endogenous
fluorescent fusion lines are available for many of the TFs analyzed in this study,
the authors should perform a small-scale comparative analysis to determine
the consistency. In addition, the authors should compare their expression
pattern to benchmark genes whose expression and function are relatively well-
documented. Although the single-cell expression at the L1 stage is not available
for most TFs, tissue-level analysis is feasible.

We thank the reviewer for this highly constructive suggestion, and we agree that the
quality of our promoter-fusion reporter profiles should be evaluated by comparison
with well-documented TFs. Previous work by Reilly and colleagues used fosmid
reporters to generate expression profiles for 101 homeodomain TFs at single cell
resolution in neurons of L4/young adults, several dozens of which showed highly
selective expression in the nervous system (Reilly *et al.*, 2020). We compared our L1
profile with the L4/young adult profile for all 84 homeobox TFs shared by two
datasets. This set included 58 promoter-fusion reporters, 22 fosmid-based reporters,
and 4 knock-in reporters (Supplementary Data 1), with fosmid and knock-in reporters
serving as gold standards because they contained the full genomic context of the
respective genes.

Target genes were then classified into four types, *i.e.*, upstream intergenic

sequence >7kb, upstream intergenic sequence <1kb, genes with >1kb intron following
the start codon, and all other genes. The cloned promoter regions of genes categorized
in the fourth group were presumed to contain the necessary and sufficient regulatory
elements for driving their expression, termed as high context promoter reporters.
Substantiating this proposition, the L1 profiles of these high-context promoter
reporters displayed strong correlations (median of ROCAUC = 0.92) with the
corresponding adult homeobox TF profiles as reported by Reilly and colleagues. This
correlation level resembled the relationship observed between the gold standard
fosmid/KI reporters and the adult profiles (median of ROCAUC = 0.90) (Figure 2C).
In our study, these closely correlated promoter reporters constituted a total of 296 TFs.
In contrast, promoter fusions belonging to the first three categories, referred to as low
context promoter reporters, displayed weak correlations with the adult stage profiles
by Reilly et al. (Figure 2C). These low-context promoter reporters encompassed 263
TFs (Supplementary Data 1).

These data have been added to lines 251 - 277 of the revised manuscript.

3. Caveats of reporter-based expression profiling. The authors should explicitly
mention and discuss the limitations of the fluorescent reporter-based strategy.
Although this approach is a reasonable choice, it has several caveats that
compromise the accuracy of the expression data. These include but are not
limited to (1) the inability to fully recapitulate endogenous expression due to
the relatively short promoter sequence included in the reporter and missing
critical regulatory elements,

As described above, we categorized the reporters of the 620 profiled TFs into three
distinct types based on the extent to which their reporter constructs encompassed
regulatory elements (Figure 2C). These categorized comprised 61 knock-in/fosmid
reporters, 296 high-context promoter reporters, and 263 low-context promoter reporters
(Supplementary Data 1). Upon conducting further analysis and comparing the results
with Reilly et al.'s adult stage homeobox TF profiles, it became evident that both knock-
in/fosmid and high-context promoter reporters effectively captured the comprehensive
expression profiles of their respective genes. In contrast, the low-context promoter
reporters demonstrated relatively weaker correlations with the published adult stage
profiles (Figure 2C).

These data have been added to lines 264 - 277 of the revised manuscript.

(2) false negatives due to low detection sensitivity of imaging,

We indeed observed this type of false negative in a protein-fusion reporter. We
generated two knock-in reporter strains of *vab-15*, including a protein fusion reporter
and a 2A::histone::mCherry co-expression reporter. The protein fusion reporter could
capture the expression dynamics of endogenous VAB-15 protein, whereas the 2A
reporter co-expressed fluorescence reporter and target gene by ribosomal skip

mechanisms during translation so that reporter protein and target protein were
generated as independent molecules. Consistent with the high stability of histone
protein, fluorescent signal of the 2A co-expression vector was observed in all cells
detected by the VAB-15 fusion reporter as well as in dozens of additional cells, such
as G2 neuroblasts (Supplementary Figure 4F). By comparison, a previously published
scRNA-seq study in worm embryos (Packer *et al.*, 2019) found that *vab-15* mRNA
could be detected at levels slightly above background in G2 neuroblasts. Moreover,
genetic analysis in the current study revealed that *vab-15* contributed to G2
proliferation potential (Figures 5F and 5L), thus confirming the endogenous
expression of *vab-15* gene in G2 neuroblasts. These collective results illustrate the
advantage of a histone::mCherry reporter in detecting weak gene expression.

These data have been added to Lines 295 - 310 of the revised manuscript.

(3) the potential influence of the integration site on expression,

Multiple worms were imaged for ninety-nine of the reporter strains, more than half of
which had correlation coefficients of $R > 0.81$ for reporter expression patterns between
individual worms of the same strain (Figure 2B). These results indicated that the nuclei
annotations were robust and, importantly, the reporter expression was reproducible. In
addition, expression profiles of 234 TF reporter constructs were obtained from multiple
worm strains (*i.e.*, with reporter integration at different sites). Comparison between
different strains showed that reporter expression patterns were less similar among
image stacks of the same reporter construct in different strains (median of $R = 0.71$)
than among different image stacks from the same strain (Figure 2B), suggesting that
differences in transgene integration site among worm strains could indeed lead to
differences in reporter expression patterns.

These data have been added to Lines 238 – 249 of the revised manuscript.

and (4) some of the expression signals are not from the subject cell but
inherited from earlier cells through the cell lineage due to long half-life of the
fluorescent protein.

This is also valid concern regarding the reliability of reporter profiles. The expression
patterns of the two above-mentioned *vab-15* knock-in reporters also differed in V5
neuroblasts (Supplementary Figure 4F). Interestingly, both reporters were positive in
Q neuroblasts, the sister cells of V5 neuroblasts. In late embryos, *vab-15* is
transcribed in the mother cells of V5 and Q neuroblasts (Packer *et al.*, 2019), and
specifies the fates of V5 and Q cells (Li *et al.*, 2017). In L1 larvae, the development of
Q neuroblasts still requires regulation by *vab-15*, while V5 development does not (Li
*et al.*, 2017). Q and V5 cells are generated by the last round of embryonic cell
division, which occurs only an hour before embryo hatching⁷. So the *vab-15*
expression observed in V5 cells of L1 larvae via the 2A co-expression reporter is

likely a false positive resulting from the long half-life of histone::mCherry reporter
protein along with the short duration from division of the mother cells to early L1
stage.

Fortunately, all other embryonic cell divisions occur more than six hours before
embryo hatching (Sulston *et. al.*, 1983), which is likely long enough to ensure that all
of the histone reporter protein is degraded. To test this hypothesis, we compared
reporter expression between daughter cells and their mother cells. In total, 126
histone::cherry reporters were profiled at the L1 stage in the current study and in
embryos up to the 350-cell stage at single cell resolution (Murray *et. al.*, 2012),
including 216 mother cells whose daughter cells were both present in L1 larvae.
Among them, 207 mother cells showed strong expression (> 30,000 intensity units) of
at least one reporter. In 129 of these mother cells, fluorescence of at least one reporter
with strong signal in the mother became undetectable in one or both daughter cells of
L1 larvae (Supplementary Data 5). Moreover, daughters of these 129 mother cells
spanned all five major tissues (intestine, muscle, neural cell, pharynx, and skin),
suggesting that histone fusion reporter proteins produced in mother cells were
completely degraded in most, if not all, somatic cells prior to worm fixation and
imaging.

These data have been added to Lines 312 - 337 of the revised manuscript.

Importantly, some of the caveats mentioned above may also affect the
reliability of TF-based cell clustering results.

We indeed observed 14 cases of discordance between TF-based cell clustering and
phenotype-based cell class classification (Figure 3). On the one hand, some differences
possibly arose from the higher resolution of TF profiling for cell type classification than
that provided by phenotyping. For example, TF-based subtyping of hypodermis cells
aligned well with the heterogeneous morphology and function between hyp 1-3 and hyp
4-11 (Figures 3, S6F and S6I). On the other hand, this distribution pointed to disparities
between phenotype-based classification and molecular clustering, specifically relying
on the profiles of all 620 TFs that were examined. Upon excluding the 263 low-context
promoter reporters from our profiles, an additional six cell classes exhibited
discordance between phenotype-based classification and TF-based clustering
(Supplementary Figure 6L). This suggests that low-context promoter reporters could
potentially offer insights into the gene regulation program sensed by specific DNA
sequences, even if they did not entirely encompass the expression patterns of their
corresponding endogenous genes.

Notably, when we excluded reporters that were likely comprehensive in capturing the
complete expression profiles of their respective genes (specifically, all 61
fosmid/knock-in reporters and a random selection of 202 high-context promoter

reporters), it is intriguing that the count of cell classes exhibiting discordance between
phenotype-based classification and TF-based clustering remained comparable to that
observed when profiles excluding the 263 low-context promoter reporters were
considered (18 vs. 20) (Supplementary Figure 6L). Moreover, we found molecular
clustering of the large set of 263 low-context reporters could recapitulate well-
established phenotype-based cell type classifications to strikingly higher extent than a
smaller set of 61 fosmid/knock-in reporters (Supplementary Figure 6K). This finding
led us to conclude that number of profiled genes affected TF-based cell type
identification more strongly than full genomic context. While we have included our
findings of low- and high-context reporter profiles in the revised manuscript, we
respectfully maintain the high number of reporter profiles for reliable TF-based cell
type annotation.

These data have been added to Lines 91 – 99 and Lines 119 – 147 of the revised
Supplementary Information.

For example, the long half-life of fluorescent protein would cause an
overestimation of the influence of cell lineage on cellular states, which should
be taken into account when interpreting the results (see below).

We indeed observed one case of influence of long half-life of fluorescent protein on
TF-based cell clustering. Q cells and V5 cells are intermingled together in the
dendrogram (Figure 3) even though they are two distinct neuroblast classes and have
distinct gene regulation programs. Q and V5 are sister cells generated by the last round
of embryonic cell division, which occurs only an hour before embryo hatching (Sulston
*et. al.*, 1983). So the failure of distinguishing these two cell classes by TF-based cell
clustering likely resulted from the long half-life of histone::mCherry reporter protein
along with the short duration from division of the mother cells to early L1 stage. But as
addressed in the answer to comment (4), all other embryonic cell divisions occur more
than six hours before embryo hatching (Sulston *et. al.*, 1983), which is likely long
enough to ensure that all of the histone reporter protein is degraded.

These data have been added to Lines 312 - 337 of the revised manuscript.

The effect of long half-life of fluorescent protein on analysis of lineage-specific gene
expression was addressed below in Major Comment #8

4. Cell clustering by TF expression. The key finding of this study is presented
in Figure 3, but the results are not well-organized, making it difficult to follow.
The use of many different colors in the figure makes it impossible to understand
the exact relationships between different cell classifications. Additionally, the
presentation of the different cell classifications in Table S4 is unclear. In
particular, organizing the classifications into a matrix is highly unintuitive. A
more effective way to present the information would be to show each cell and
then different clustering or classifications as numbers or text.

We are very grateful for the reviewer's advice. Figure 3 was split to Figure 3 and Figure
4 to enlarge symbols and characters. Secondly, cell classes were labeled by numbers to
facilitate visual inspection in Figure 3. At last, Table S4 has been revised as
recommended and renamed as Supplementary Data 6.

5. Gene-level analysis. The cellular expression patterns of individual TFs are
informative in revealing regulators of specific cell fates, types, or states.
However, the authors did not perform any analysis to characterize TFs with
interesting yet uncharacterized expression patterns. At the very least, the
authors should identify and discuss individual TFs exhibiting cell-type or cell-
state-specific expression patterns. This analysis is critical because it helps to
demonstrate the value of high-resolution expression analysis to identify novel
expression enrichment/functions of TFs. Furthermore, it is also an integral part
of quality control to show how many known patterns are recapitulated (see
Major comment 2).

We appreciate these highly constructive suggestions. In revised manuscript, we detailed
absolutely class-specific TFs, which were defined as genes expressing in all members
of one cell class, but not in any cell of other cell classes. 12 TFs showed absolute
specificity in seven cell classes, including six TFs (*ada-2*, *elt-2*, *ets-9*, *nhr-121*, *nhr-176*,
and R09H10.3) in the intestine, *pes-1* in the Z1/Z4 somatic gonad precursor, *npax-4* in
the phasmid sheath, *ceh-44* in IL2 neurons, *pqm-1* in LUA neurons, and *che-1* and *nhr-*
*74* in amphid neurons ASE and AWA, respectively (Supplementary Data 4). The large
number of intestine-specific TFs supports the notion that endoderm is a special tissue
in the transcriptome (Hashimshony *et al.*, 2015). Our profiles included five known
intestine regulators (*elt-2*, *elt-7*, *end-1*, *end-3*, and *med-1*) (Cissy Yu *et al.*, 2015). All
but *med-1* were expressed in all 20 intestinal cells in our profiling. Unlike intestine, no
specific TF was detected in bodywall muscles in our profiles. Bodywall muscles are
specified by three redundant master regulators, *hlh-1*, *unc-120* and *hnd-1* (Fukushige *et*
*al.*, 2006). Each of these regulators is expressed in multiple cell types. Nevertheless,
two major master regulators, *hlh-1* and *unc-120*, were co-expressed exclusively in all
81 bodywall muscles (Supplementary Data 2), distinguishing them molecularly from
all other cell classes.

These data have been added to Lines 348 – 365 of the revised manuscript.

Additionally, we compared our L1 profile with the L4/young adult expression data for
84 homoeobox TFs profiled by fosmid reporters (Reilly *et al.*, 2020). Results are
described in the answer to Major comment 2.

6. Quiescent HSN state. TF expression status in the sister cell of HSNs (PHBs)
should be included as an internal control to demonstrate gene repression in the
quiescent HSN state. As the authors use the fluorescent signal to monitor gene
transcription, protein turnover is likely involved in addition to repressing gene
expression, and this possibility should be discussed. Moreover, if possible, the

authors should use more endogenously-tagging reporters to verify the
proposed gene repression in HSN neurons.

We agree that these controls are critical for our hypothesis on the HSN quiescence state.
As recommended, we included PHB as an internal control in revised manuscript
(Figures 4G and 4H, and Supplementary Figures 5J-5M).

Moreover, our updated profiles included 3 protein fusion reporters (*ceh-38*, *lin-13*, *snu-*
*23*) whose expression was detected in nearly all neurons. The expression level of each
of these reporter in HSN class was the bottom-2 among all 96 neuron classes
(Supplementary Figure 4G). Altogether, these results support our hypothesis that TF
expression is repressed in HSNs at the L1 stage.

These data have been added to Lines 417 – 420 of the revised manuscript.

Furthermore, since HSNs are hermaphrodite-specific neurons (the equivalent
cells undergo programmed cell death in males), it is worth discussing whether
apoptosis-related pathways play a role in gene repression.

In revised manuscript, we examined expression patterns of *saeg-2* *mosSCI* and *sma-9*
knock-in reporters in *ced-3* mutants. Gene expression in HSNs remained repressed in
this apoptosis-defective mutant at the L1 stage (Supplementary Figure 5J-5M).

These data have been added to Lines 441 – 443 of the revised manuscript.

7. P and G2/W states. This section is difficult to follow, and it is unclear what
the authors intended to highlight. If I understand correctly, the authors
identified P and G2/W as two subtypes of neuroblasts. They then conducted
follow-up experiments and characterized the shared and distinct molecular
regulation of the two states by several known regulators (*ref-2*, *vab-15*, and
*tlp-1*). At first glance, the authors seem to highlight the molecular distinction
of the subtypes.

Sorry for the ambiguity in our description. Through characterization of the molecular
distinction of neuroblast subtypes, we want to highlight that whether two subtypes share
the same pro-neural battery can be determined by stratification of differential TF
expression related to their morphological heterogeneity.

However, the authors concluded the findings as "... revealed a TF battery (*ref-*
*2*, *vab-15*, and *tlp-1*) shared between the TF-based G2/W and P neuroblast
subtypes.". In addition, it is unclear what hypothesis the authors are trying to
form in this long sentence: "So we speculated that two neuroblast subtypes
specified by similar pro-neural TF batteries would have smaller inter-subtype
difference in TF expression normalized by their cross-sister clade difference
than that between subtypes specified by different TF batteries (Supplementary
Figure 5D)." This section should be significantly revised to make it clear what
the hypothesis is and how each finding supports it.

Our rationale is that whether two neuroblast subtypes are specified by the same pro-
neural TF battery can be determined by excluding intra-subtype differences in TF
profiles related to morphological heterogeneity. It is reasonable to first assume that
different TF batteries underlie morphological varieties of epithelial subtypes. Then
neuroblasts can be considered as special epithelial cells expressing proneural TF
batteries. Moreover, different pro-neural TF batteries underlie various neural
developmental programs. As such, differences in whole TF profiles between neuroblast
subtypes are composed of differences in their morphological TF batteries and pro-
neural TF batteries. For neuroblast subtypes specified by the same pro-neural battery,
differences in their complete TF profiles still can be large if they have dramatically
different morphological TF batteries. If we manage to exclude differences in
morphological TF batteries, molecular similarity between neuroblast subtypes driven
by the same pro-neural TF battery can be revealed. Excluding the difference in
morphological TF batteries was computationally achieved by normalization of their
cross-neighbor clade difference in JSD score (Supplementary Figure 6D).

These data have been added to Lines 454 – 462 of the revised manuscript and Lines
201 – 214 of the revised Supplementary Information.

8. Lineage-dependent heterogeneity. As mentioned in Major Comment 3, the
long half-life of fluorescent proteins will make early expressed proteins persist
in descendants through the cell lineage even though the endogenous
transcripts or proteins no longer exist. As a result, the lineage effect may be
overestimated since the propagation of protein is through mitotic divisions in
each lineage. To address this concern, the authors should validate their findings
by focusing on reporters that are expressed since the L1 stage, thereby
avoiding this caveat.

We agree that excluding the effect of long half-life of fluorescent proteins is critical to
address the effect of embryonic cell lineage on post-embryonic gene expression and
cell phenotypes. However, development from L1 to young adult takes about two days,
which is much longer than the half-life of histone::mCherry reporter protein as
addressed in Comment 3(4). For example, histone::mCherry reporters of six TFs (*med-*
*1*, *hlh-2*, T22C8.4, *zip-7*, *tbx-8*, and *tbx-9*) were expressed in MS-derived bodywall
muscles of L1 larvae, but became undetectable in those of young adults (Figures 6B
and 6C). Moreover, while *ref-1* was specifically expressed in the right RMDD neuron
of L1 larvae (Figure 6D), its expression was undetectable in the neuron of late embryos
(Murray *et al.*, 2008; Ma *et al.*, 2021), suggesting post-embryonic initiation of
asymmetric expression of *ref-1*.

At last, the asymmetric role of *hnd-1* in post-embryonic knock-out experiments also
excluded the explanation of long half-life histone::cherry protein for *hnd-1* gene. These
collective results illustrate the effect of embryonic cell lineage on post-embryonic gene
expression and cell phenotypes.

These data have been added to Lines 524 - 533 and 578 – 589 of the revised manuscript.

Minor Comments:

Line 83: “Moreover, a significant source of this intra-type molecular
heterogeneity was convergence of different lineages into the same cell fate, i.e.,

multiple cell lineages could produce cells of the same phenotype and function.”

– A complete convergence should not produce heterogeneity.

Sorry for our ambiguous statement. Our statement emphasizes on convergence at
phenotypic level. So the sentence has been rephrased to:

“Moreover, a significant source of this intra-type molecular heterogeneity was

convergence of different lineages into the same **phenotypic cell type**, i.e., multiple cell

lineages could produce cells of the same **morphology** and function.”

These data have been added to Lines 79 – 80 of the revised manuscript.

Line 207-211: Is the integration site the only difference between these strains?

It’s a good point. Our original manuscript did not distinguish different types of reporters.

In revised manuscript, this part addressed the 234 promoter reporter constructs have

multiple strains profiled. Major difference between these strains is the integration,

although copy number variance in bombardment strains might have minor influence.

These data have been added to Lines 242 – 249 of the revised manuscript.

site Line 218: The correlation coefficient should be provided instead of being
displayed as a color gradient.

done as recommended (Figure 2D)

Line 225: Figure 3E-G should be Figure 2E-G, also Figure 2H is not cited before
Figure 3

Sorry for our neglect. It has been revised (Line 291). The original Figure 2H becomes

Figure 4F in the revised manuscript.

Line 257: sister-clade cells, To avoid confusion with sister cells, I would suggest
using a term like "neighbor-clade cells" or something similar

done as recommended

Lin 1190-1192: “The mean expression profile across all image stacks of a strain
was used as the strain expression profile. The mean expression profile across

all strains of a TF was used as expression profile of the TF.” – While it is

reasonable to average expression across experimental replicates, it is uncertain

whether it is valid to average expression between different reporters for the

same transcription factor, especially if the expression levels differ significantly

We are very grateful for the reviewer’s advice. We had shown that different types of

reporters captured the full expression of the respective genes to different extent (Figure

2C). From high to low were fosmid/knockin, high-context promoter, and low-context
promoter reporters. It is reasonable that the final expression profile should be based
on reporters capturing the most comprehensive expression of the respective genes. So
in revised manuscript, if there were different types of reporters for the same TF, the
reporter type with higher faithfulness was chosen.

These data have been added to Lines 823 – 827 of the revised manuscript.

Figure 3A: Rather than organizing the hierarchical clustering into a circular-
shaped plot, a classic horizontal view would be more intuitive and effective.
Additionally, the current plot does not allow for easy visualization of which cells
belong to the same clade

We agree with the reviewer at this point. A horizontal view was indeed our first attempt.
But it claims too much space. So we keep using a circular dendrogram.

Line 324: in the paragraph, Figure 6 should be Figure 5.

Sorry for our neglect. It has been revised (Lines 464 - 475).

Line 370: "It is well-established that 97 of these 159 cell pairs are symmetric
in cell lineage, while the other 62 pairs are converged cell pairs, whose left and
right cell mates have an asymmetric cell lineage history." –To clarify the
meaning of "symmetric" and "asymmetric," an example should be provided.

In revised manuscript, two examples (vir and im) were provided, both of which showed
asymmetric gene expression (Figure 6D) in our profiling.

These data have been added to Lines 513 – 517 of the revised manuscript.

We would like to thank Reviewer #3 for careful reading of our paper, and for his highly
constructive comments, which have helped us to improve our analyses and ultimately
strengthen our conclusions.

REVIEWERS' COMMENTS

Reviewer #1 (Remarks to the Author):

The authors have done a very good job in addressing my concerns and I can now recommend publication as is.

Reviewer #2 (Remarks to the Author):

The authors have addressed my concerns. While not emphasized in my original review, I completely agree with the other reviewers in terms of the value of the study to the field.

Reviewer #3 (Remarks to the Author):

The revised manuscript has undergone substantial improvements, and all of my concerns have been adequately addressed. It is now well-suited for publication in Nature Communications.